# RAS-independent ERK activation by constitutively active KSR3 in non-chordate metazoa

Aline Chessel[1], Noémie De Crozé[1], Maria Dolores Molina[2], Laura Taberner [1], Philippe Dru [3], Luc Martin[1] & Thierry Lepage [1] ✉

During early development of the sea urchin embryo, activation of ERK signalling in mesodermal precursors is not triggered by extracellular RTK ligands but by a cell-autonomous, RAS-independent mechanism that was not understood. We discovered that in these cells, ERK signalling is activated through the transcriptional activation of a gene encoding a protein related to Kinase Suppressor of Ras, that we named KSR3. KSR3 belongs to a family of catalytically inactive allosteric activators of RAF. Phylogenetic analysis revealed that genes encoding kinase defective KSR3 proteins are present in most non-chordate metazoa but have been lost in flies and nematodes. We show that the structure of KSR3 factors resembles that of several oncogenic human RAF mutants and that KSR3 from echinoderms, cnidarians and hemichordates activate ERK signalling independently of RAS when overexpressed in cultured cells. Finally, we used the sequence of KSR3 factors to identify activating mutations of human B-RAF. These findings reveal key functions for this family of factors as activators of RAF in RAS-independent ERK signalling in invertebrates. They have implications on the evolution of the ERK signalling pathway and suggest a mechanism for its co-option in the course of evolution.

The RAS-RAF-MEK-ERK MAPK pathway is one of the most conserved signalling cascades that link the response of cells to external signals in eukaryotes[1]. In metazoans, the ERK MAPK pathway plays crucial roles both during embryonic development and in homeostasis in the adult, orchestrating fundamental processes such as cell fate specification, proliferation, migration, differentiation, apoptosis and metabolism[2,3]. In humans, the ERK pathway is also strongly involved in diseases and mutations in several components of the pathway are responsible for a large proportion of cancers and disorders known as RASopathies[4,5]. Although RAF plays crucial roles in essential processes in animals and is a pivotal factor in tumorigenesis in humans, the molecular mechanisms regulating its activation are not completely understood and the very early steps of the RAF activation cycle remain enigmatic[6]. Activation of RAF critically relies on dimerization of RAF/RAF or RAF/

KSR dimers and involves a highly conserved arginine residue in the dimerization interface[7]. Dimerization is thought to induce a transactivation process in which one of the protomers, called the activator, allosterically activates its partner, called the receiver[8–10]. Allosteric activation of RAF is accompanied by changes in its phosphorylation state but it does not require the kinase activity of the activator protomer. Activation of RAF requires instead phosphorylation of a motif on the activator called the NtA motif, located at the N-terminus of the kinase domain. The exact role of this phosphorylation is not fully understood but it has been shown that the NtA motif engages in interprotomer salt bridges that stabilize the binding interface between two B-RAF protomers[11].

The established model in the field of ERK signalling is that this pathway is activated by extracellular ligands binding to Receptor

[1]Institut de Biologie Valrose CNRS, Université Côte d'Azur, Nice, France. [2]Department of Genetics, Microbiology and Statistics, Faculty of Biology, University of Barcelona, Barcelona, Catalonia, Spain. [3]CNRS, Laboratoire de Biologie du Développement de Villefranche-sur-Mer (LBDV), Institut de la Mer de Villefranche, 181 Chemin du Lazaret, 06230 Villefranche-sur-Mer, France. ✉e-mail: tlepage@unice.fr

Tyrosine Kinases (RTKs), which in turn relay the external information to the cytoplasm of the cell through RAS. Indeed, besides cases of oncogenic activation of RAF, which result in RAS-independent MEK/ERK signalling, and with the very few exceptions such as MEK activation by the regulated translation of MOS mRNA during oocyte maturation in amphibians, there is no reported example of activation of this pathway independently of RAS and constitutively active allosteric regulators of RAF have never been described.

The process of skeletogenesis in the early sea urchin embryo offers one such intriguing example of a cell-autonomous, RAS-independent ERK activation[12,13]. The skeleton of the sea urchin embryo is built from the Primary Mesenchymal Cells (PMCs), a population of mesenchymal cells that delaminates from the vegetal pole region of the embryo at the blastula stage. The Gene Regulatory Network (GRN) controlling specification of the PMCs has been studied in depth in several species of sea urchin[14,15]. In euechinoids, which include most extent urchins except cidaroids (pencil urchins), specification of the PMCs is initiated by the activity of Pmar1, a homeobox transcriptional repressor that is expressed in the progeny of the micromeres downstream of maternal beta-catenin[16]. The current model is that Pmar1 acts as a micromere specific repressor of the expression of HesC, a ubiquitously expressed repressor of the PMC fate that belongs to the Hairy Enhancer of Split-C family of repressors[17] (Fig. 1a). This double repression mechanism controls the PMC-specific expression of genes encoding key transcription factors such as Alx1 and Ets1, which are major and early actors of the PMC GRN[18–20]. Overexpression of *pmar1* converts any cell of the embryo into a PMC thereby triggering a massive epithelial-mesenchymal transition (EMT) in the embryo, which is entirely transformed into a ball of migrating mesenchymal cells. However, this mode of skeletogenesis, which relies on micromeres and on the Pmar1/HesC double negative gate, is considered as a recent innovation in the evolutionary history of echinoderms. The double repression circuit involving Pmar1 and HesC is absent in ancestral urchins such as the cidaroids, which make an embryonic skeleton not from micromeres but from mesenchymal cells that bud off from the tip of the archenteron during gastrulation. The most likely scenario is therefore that the double negative gate circuit has been co-opted in euechinoids, a process that required to express *pmar1* downstream of beta catenin in the micromere lineage and to place the early acting genes of the GRN under the repression of HesC.

In euechinoids, the ERK pathway plays a crucial role in the GRN that controls formation of the skeleton. When the ERK pathway is inhibited, the PMC GRN collapses and expression of key regulatory genes such as *alx1* and *tbr* vanishes, resulting in embryos that fail to form PMCs and that completely lack a skeleton[12,13]. Blocking the pathway at the level of either ERK, MEK or RAF blocks specification of the PMCs by inhibiting phosphorylation of ETS1 and produces PMC deficient blastulae and gastrulae lacking a skeleton. Intriguingly, however, blocking the pathway at the level of RAS or suppressing cell-cell interactions does not prevent the formation of PMCs suggesting that an alternative RAS-independent mechanism of activation of ERK is operating in this process[12]. Since 2004, attempts to identify the mechanism responsible for the cell-autonomous ERK activation failed and we still ignore the nature of the signal that activates ERK and the position of the gene that encodes this signal in the PMC GRN.

In this work, using gain and loss-of-function approaches we first identified the *pmar1/hesC* as the genetic circuit that operates immediately upstream of ERK activation in the sea urchin embryo. Using transcriptome profiling we searched for genes that are induced by Pmar1 and identified a transcript encoding a member of a previously undescribed family of Kinase Suppressor of Ras (KSR) proteins as the key message whose accumulation triggers ERK activation in the PMCs downstream of Pmar1. Importantly, expression of this *ksr3* transcript was found to be necessary and sufficient for cell-autonomous and RAS-

independent activation of the ERK pathway. We identified orthologs of this *ksr3* gene in most non-chordate metazoa and showed that several of them also activate ERK signalling independently of RAS. These data reveal that unlike the previously identified KSRs, which act exclusively in RAS-dependent signalling, KSR3 factors act as potent RAS-independent activators of RAF that trigger cell-autonomous activation of ERK. Our results allow to identify a previously unknown branch of the ERK signalling pathway, provide insights into the mechanism of activation of RAF and have implications on the evolution of this key signalling pathway and on the process of co-option of signalling pathways in the course of evolution.

## Results

### Activation of ERK signalling in mesodermal precursors is controlled by Pmar1

Activation of ERK in precursors of the skeletogenic mesoderm of the sea urchin embryo requires the activity of βCatenin/TCF but the genetic circuit regulating generation of the signal that activates ERK is not known (Fig. 1a)[12]. As expected from the RAS-independence of the EMT of PMC precursors previously observed, activation of ERK at the vegetal pole occurred normally in embryos injected with mRNA encoding a dominant negative RAS factor (Fig. 1b). However, the function of RAF is likely required in this mechanism since overexpression of a dominant negative form of RAF blocked activation of ERK at the vegetal pole and abolished formation of the PMCs (Fig. 1b). To identify the sub-circuit of the PMC GRN that regulates activation of ERK, we first compared the kinetics of ERK activation with the kinetics of zygotic expression of Ets1 and Alx1, the two key transcriptional regulators of the PMC GRN (Fig. 1a, c). Strong phospho-ERK immunostaining was detected at the vegetal pole of embryos as early as at the very early blastula stage, i.e. well before the prehatching blastula stage that we previously considered as the initial stage of ERK activation[12]. The precocious timing of activation of ERK signalling was consistent with a model in which the signal that activates ERK is produced downstream of early expressed factors such as Pmar1, Alx1 or Ets1 factors. However, although injections of antisense morpholino oligonucleotides targeting *ets1* or *alx1* transcripts[19] prevented formation of the PMCs, blocked differentiation of the skeleton and abrogated expression of PMC marker genes such as *dri* and *msp130* (Fig. 1d), they did not block ERK activation in mesodermal precursors at blastula stage, eliminating *ets1* and *alx1* as candidates genes promoting ERK activation. Furthermore, overexpression of *ets1* (Fig. 1e) failed to induce ectopic activation of ERK signalling, strongly suggesting that ERK activation does not require the activity of this transcription factor and consistent with previous findings showing that MAPK signalling is crucial for the activity of Ets1 and for the expression of *alx1*[12]. In contrast, overexpression of *pmar1*, which acts upstream of *ets1* and *alx1* induced a massive activation of ERK throughout the embryo at early blastula stage, followed by a strong ectopic expression of PMC marker genes such as *dri* and by a massive EMT. This result indicated that Pmar1, but not Ets1 or Alx1, acts immediately upstream of the Ras-independent activation of ERK signalling in the PMCs.

### An RNA-seq screen identifies kinase suppressor of Ras (KSR) as a candidate activator of ERK signalling downstream of Pmar1

Having established that activation of ERK in mesodermal precursors requires the activity of the transcription factor Pmar1, we attempted to identify the "signal" that activates ERK by comparing the transcriptome of early embryos overexpressing *pmar1* to that of embryos lacking the activity of Pmar1. Since the function of Pmar1 cannot be easily blocked using morpholinos due to the expression of multiple transcripts isoforms generated from the *pmar1* locus, we used instead as a proxy for the Pmar1 loss-of-function condition embryos animalized by overexpression of a dominant negative cadherin construct, a

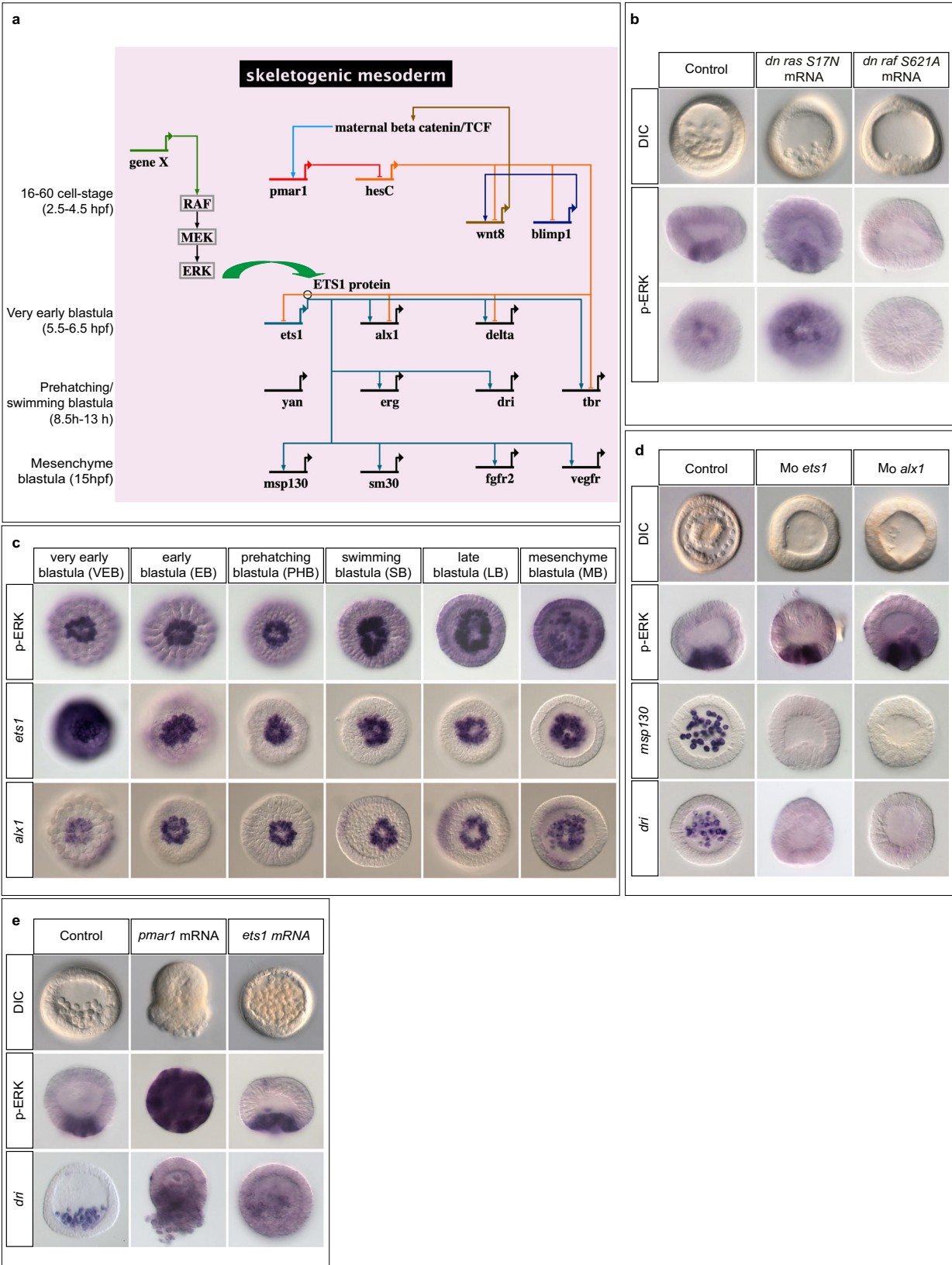

perturbation which inhibits the beta-Catenin pathway, blocks *pmar1* expression and suppresses PMC specification and ERK activation (Fig. 2a)[21,22]. A cursory review through the list of candidate genes isolated in this screen indicated that it successfully recovered most of the known early players of the PMC GRN including *alx1, alx4, ets1, kruppel-like, wnt8, wnt1, Delta* and *blimp1* (Fig. 2b). One transcript recovered in

the RNA profiling experiment was annotated as similar to the serine/threonine-protein kinase-transforming protein Rmil, the activated product (deleted of the N-terminal domain) of the chicken *b-raf* proto-oncogene, which plays a central role in the activation of the ERK pathway[5,23]. A closer look at the sequence of this transcript indeed revealed that it was not the product of the *raf* gene, but the product of

**Fig. 1 | The ERK pathway is activated early in the skeletogenic precursors downstream of *pmar1* but independently of *alx1* or *ets1*. a** Biotapestry[82] diagram of the simplified gene regulatory network (GRN) driving specification and differentiation of the skeletogenic mesoderm in *Paracentrotus lividus* (see also[14, 15, 20]). A time scale is given from top to bottom on the left side. The GRN is activated by nuclearization of maternal beta-catenin in micromeres around the 16-cell stage. In the micromeres, beta-catenin triggers the expression of the homeobox gene *pmar1*. By repressing the expression of the ubiquitously expressed repressor of the PMC differentiation program, *hesC*, in the PMCs, Pmar1 allows the expression of regulatory genes further downstream in the GRN including *wnt8*, *blimp1*, *ets1*, *alx1*, *erg*, *yan*, *tbr* and *dri* in the skeletogenic mesoderm. Ets1 is a central regulator of this GRN and regulates the expression of two layers of genes encoding transcription factors including *aristaless-like1* (*alx1*), *tbrain* (*tbr*), *yan/tel*, *erg* and *dead ringer* (*dri*)[12, 13]. Activation of the RAF/MEK/ERK pathway is crucial for the activity of Ets1 and for the expression of *alx1* but the signals/genes responsible for activation of the pathway and their exact position in the GRN are not known (gene X). The green arrow depicts phosphorylation of Ets1 by ERK. **b** Activation of ERK in the PMC precursors occurs independently of RAS but requires RAF. **c** time-course of ERK activation and of *ets1* and *alx1* expression reveals that activation of ERK in the PMCs coincides with the onset of zygotic expression of *ets1* and *alx1* in these cells. The strong in situ hybridization signal for *ets1* at very early blastula stage is due to maternal *ets1* transcripts. Activation of ERK is detected by immunostaining with an antibody against the di-phosphorylated (activated) form of ERK. All the embryos in (**c**) are viewed from the vegetal pole. Note that at very early blastula stage, which corresponds to the beginning of ERK activation, only 50 % of the embryos show ERK activation in the PMC precursors while 100% of the embryos are labelled at later stages. **d** Morpholino mediated knock-down of *ets1* or *alx1* blocks delamination of the PMCs and suppresses expression of PMC marker genes *msp130* and *dead ringer* (*dri*) but it does not inhibit activation of ERK in the PMCs. **e** although overexpression of either *ets1* or *pmar1* can trigger a massive EMT at blastula stage, only overexpression of *pmar1* induces massive ectopic activation of ERK. The number of times the experiments were replicated and the number of embryos analyzed are indicated in See Supplementary Table 1.

a gene encoding a protein related to RAF named Kinase Suppressor of RAS (KSR). Kinase suppressor of Ras proteins were identified as positive regulators of ERK signalling downstream of RAS signalling in genetic screens in the fly and in *C. elegans*[24–26] and genetic epistasis experiments suggested that, in *Drosophila*, KSR was acting between RAS and RAF[24–26]. Furthermore, in vertebrates, KSR proteins have been proposed to act as important scaffold proteins that coordinate the assembly of RAF, MEK and ERK[27] and as allosteric activators of RAF[7,8,10,28,29]. The *ksr* gene identified in the screen therefore appeared as a remarkably attractive candidate for the factor that triggers the RAS-independent activation of ERK signalling in the PMC precursors of the sea urchin embryo. We analysed the complement of *raf/ksr* genes present in the sea urchin genome and found that, in addition to this *ksr*-like sequence, it contains a single prototypical *raf* gene and a second gene encoding a KSR protein mostly related to KSR1 and KSR2 from vertebrates. We therefore named the novel *ksr* sequence *ksr3* (Fig. 2b, c). Although the sea urchin KSR1 protein shared many features with the KSR proteins from vertebrates, flies and nematodes, the structure of the protein encoded by the sea urchin *ksr3* transcript isolated in the screen differed radically from that of any known KSR strongly suggesting that this *ksr3* gene was not orthologous to any previously characterized *ksr* gene.

### KSR3 lacks the N-terminal CC-SAM domain and carries a highly remodelled ATP-binding site and catalytic loop

The structure of sea urchin KSR3 differs markedly from that of prototypical *Drosophila* or vertebrate KSR proteins strongly suggesting that *ksr3* is not the ortholog of any of these *ksr* genes (Fig. 2c). In particular, the sea urchin KSR3 protein lacks the proline-rich domain and the N-terminal coiled-coil sterile alpha motif (CC-SAM domain) that is present in the sea urchin KSR1 protein as well as in the fly and vertebrate KSR proteins[28]. The sea urchin KSR3 protein also differs significantly from the prototypical KSR proteins at the level of the kinase domain. KSR proteins from *Drosophila* or *C. elegans* display many conserved features of prototypical kinases[30] including the consensus Gly-X-Gly-X-X-Gly, which stabilizes the non-transferable phosphate groups of ATP during binding of ATP to the enzyme, the DFG motif at the beginning of the activation loop that chelates the divalent magnesium cation and stabilizes the β- and γ-phosphate groups of ATP, or the VAV/I K of subdomain II that contains the lysine (K483 in human BRAF) that plays an essential role in the phospho- transfer reaction[5]. In contrast, the sea urchin KSR3 protein contains a modified Glu-X-Gly-X-X-Gly motif and the highly conserved DFG motif is absent (Fig. 2c), (Supplementary Fig. 2). Similarly, although the sea urchin KSR1 contained a canonical VAIK motif like its fly and *C. elegans* counterparts[7], the sea urchin KSR3 protein contains a highly divergent VLIQ motif in place of the

canonical VAVK or VAIK motifs (Supplementary Fig. 2). Finally, the HRD motif of the catalytic loop that contains the catalytic aspartate (D576 in BRAF) and that is highly conserved in all kinases, is replaced by a partially conserved HKD in KSR1 whereas it is simply absent in the sea urchin KSR3 protein (Supplementary Fig. 3). The highly remodelled ATP binding pocket and the presence of divergent activation segment and catalytic loop strongly suggest that KSR3 is a non-catalytic kinase.

### *ksr3* is expressed dynamically in all three germ layers

As expected for a target of Pmar1, the early *ksr3* expression closely followed that of *pmar1*: its mRNA was not present in the egg, it started to accumulate in the skeletogenic mesoderm at the 60-cell stage, its level peaked at the swimming blastula stage, then, after ingression of the PMCs, the transcript level decreased significantly in the skeletogenic mesoderm and at the beginning of gastrulation only residual *ksr3* expression was detected in the PMCs (Fig. 3a, b). At the early gastrula stage, novel territories of *ksr3* expression appeared at the vegetal pole region in the primary invagination of the archenteron, then in 3–5 single cells located at the animal pole region. At the late gastrula stage, *ksr3* was expressed in the endoderm, in the hindgut region and in the roof of the archenteron. Notably, it was also expressed in the overlying apical plate ectodermal region, in 4 to 6 cells, which, based on their number and location, likely correspond to serotonergic neurons[31]. At the prism stage, *ksr3* continued to be expressed in single cells of the animal plate as well as in one cell located on the left side of the archenteron. At the early pluteus stage, *ksr3* continued to be expressed in a row of contiguous cells in the apical plate, as well as in discrete cells of the ciliary band ectoderm and in pairs of ectodermal cells at the tip of the post oral arms. Finally, in 3-week old pluteus larvae, *ksr3* was expressed in the adult rudiment, in the developing appendages such as the tube feet and spines.

### *ksr3* expression pattern overlaps with domains of ERK activation

Anti-phospho ERK immunostaining revealed that the territories expressing *ksr3* overlap with domains of ERK activation (Fig. 3c). Phospho-ERK was detected in the early invaginating archenteron and in the animal plate at the beginning of gastrulation, and at the prism stage, in one cell on the left side of the archenteron, in single cells of the ciliary band and in the apical plate. Finally, phospho-ERK was detected in the apical plate and in cells located at the tip of the post oral arms at pluteus stage. Therefore, both the early and late pattern of expression of *ksr3* coincide with domains of ERK activation, both in the skeletogenic mesoderm, in the endoderm and in neural territories. In contrast, *ksr1* expression was detected from the unfertilized egg up to pluteus stage and *ksr1* messages were present at a low level and ubiquitously (Fig. 3d).

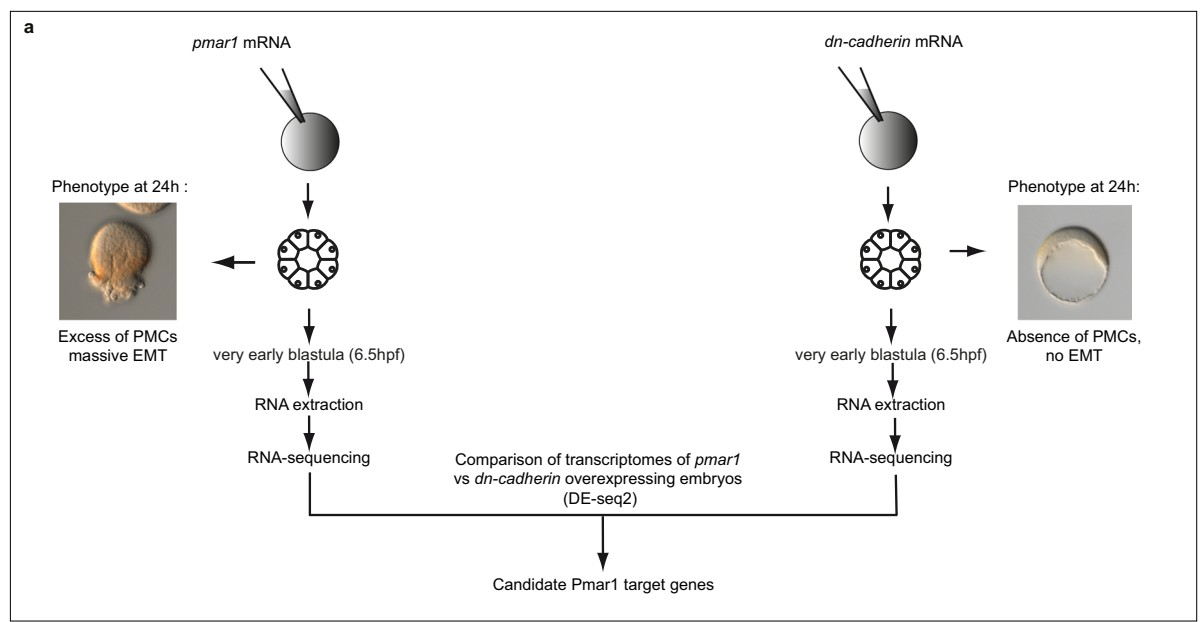

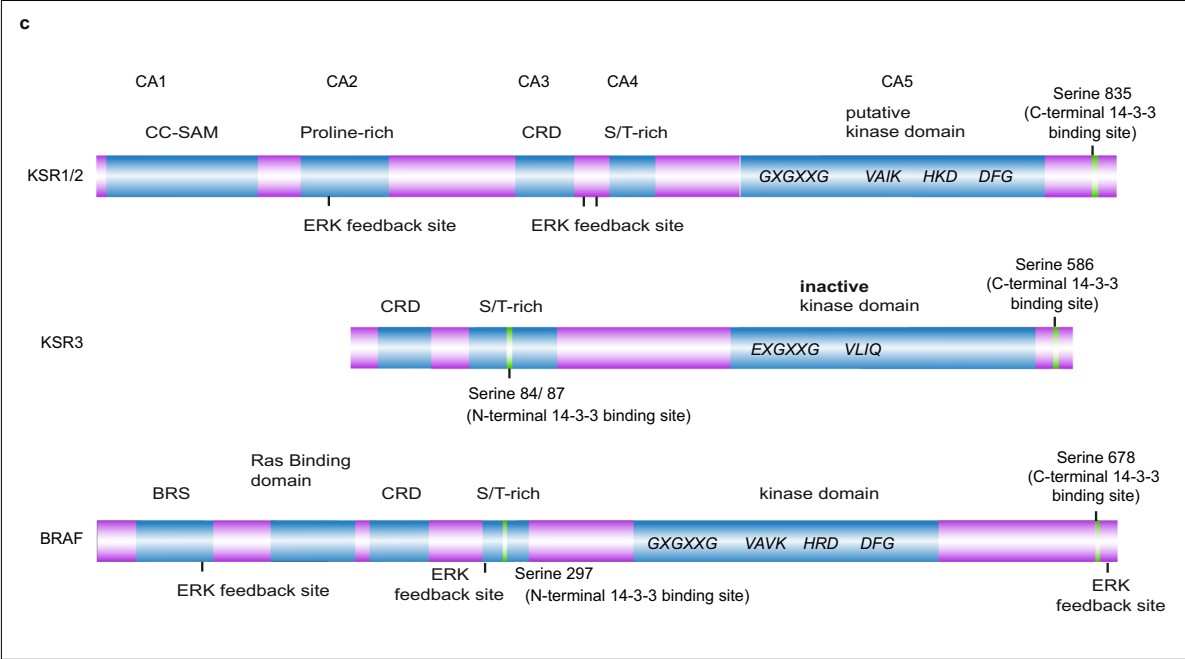

| id P.lividus | Log2FC | FDR | Spu_name | *S. purpuratus* description | Swissprot_hit |
|---|---|---|---|---|---|
| Pliv14243 | 5,63 | 0,00 | Z474 | zinc finger protein 474, Sp -Z13, transcriptional repressor Krl (SpKrl)-like | CHE1_CAEEL |
| Pliv11320 | 5,49 | 0,00 | Alx1 | aristaless-like homeobox 1 -like, Alx1(Cart1)/Alx3/Alx4 subfamily -like, Sp-Alx1_1 | ALX4_HUMAN |
| Pliv03381 | 5,46 | 0,00 | Ets1 | v-ets erythroblastosis virus E26 oncogene homolog 1/2 -like , ets1 | ETS1_STRPU |
| Pliv19212 | 5,41 | 0,00 | Wnt1 | wingless-type MMTV integration site family, member 1 | WNT1_STRPU |
| Pliv04437 | 4,92 | 0,00 | Delta | Delta (Dl) homolog, neurogenic locus protein delta, Dll1/4 -like | DL_DROME |
| Pliv27756 | 4,76 | 0,00 | **Ksr 3** | kinase suppressor of Ras 2 | KSR2_MOUSE |
| Pliv22106 | 4,35 | 0,00 | Wnt8 | wingless-type MMTV integration site family, member 8 | WNT8_XENLA |
| Pliv17449 | 2,66 | 0,01 | Blimp1 | B lymphocyte induced maturation protein -like, PRDM1-like, blimp1/krox, SpKrox1 | Blimp1_STRPU |
| Pliv19216 | 1,64 | 0,02 | Tbr1 | T-box brain transcription factor 1 | Tbr1_STRPU |
| Pliv28067 | 1,69 | 0,03 | Dri | Protein dead ringer homolog | Dri_STRPU |

## KSR3 is required for activation of ERK signalling in the precursors of the PMCs

Overexpression of *pmar1* by injection of mRNA into fertilized eggs induced a massive ectopic expression of *ksr3* detectable by in situ hybridization at blastula stage, followed by a global EMT at the beginning of gastrulation (Fig. 4a). Since Pmar1 acts by repressing the

expression of *hesC* we tested the effect on *ksr3* expression of injection of morpholino oligonucleotides targeting *hesC*[17]. Like overexpression of *pmar1*, injection of morpholino oligonucleotides targeting the *hesC* transcript caused strong ectopic activation of ERK at blastula stage, followed by a massive EMT at the beginning of gastrulation supporting the idea that Pmar1 drives expression of the *ksr3* gene in the PMCs by

**Fig. 2 | Transcriptome profiling identifies Kinase Suppressor of Ras 3 as a candidate for activation of ERK in the skeletogenic precursors. a** Design of the screen. *pmar1* overexpression induces a massive activation of ERK and causes formation of an excess of PMCs while overexpression of *DN-cadherin* prevents expression of *pmar1*, blocks activation of ERK, suppresses formation of PMCs and eliminates expression of vegetal pole marker genes by sequestering β-catenin at the level of the membrane. RNA profiling of embryos overexpressing *pmar1* or *dn-cadherin* should therefore allow to identify the gene(s) that activates ERK signalling. **b** Top list of positive controls and novel genes recovered in the screen for genes upregulated following overexpression of *pmar1*. The identification numbers of the *Paracentrotus lividus* gene predictions are given in the first column, together with the $Log_2$ (fold change) between the *pmar1* overexpression versus dn-*cadherin* overexpression conditions. FDR, false discovery rate. Spu, *Strongylocentrotus*

*purpuratus*. All the genes involved in activation of the PMC GRN in different species were present in the list of candidates genes upregulated in *pmar1* overexpressing embryos compared to DN-*cadherin* overexpressing embryos. The only gene related to MAPK/ERK signaling in this list of candidates encoded a member of the Kinase Suppressor of Ras family: KSR3. **c** Domain organization of the sea urchin KSR1, KSR3 and BRAF. KSRs and RAF proteins share conserved domains such as the kinase domain, the S/T rich domain and the cysteine rich domain or CRD (also called C1 domain). In contrast the BRAF-specific domain (BRS) and the RAS binding domain (RBD) are uniquely present in BRAF while the CC-SAM domain is present in the sea urchin KSR1 but absent in KSR3. Several motifs highly conserved in kinases are also missing or highly divergent in KSR3 including the GXGXXG and VAV/IK motif of the ATP binding site, the HRD motif of the catalytic loop, and the DFG motif of the activation loop.

repressing *hesC* expression (Fig. 4a). However, we noted that the phenotype caused by inhibition of *hesC* was consistently less severe than the phenotype caused by overexpression of *pmar1*. In contrast, overexpression of mRNA encoding the *hesC* repressor, or encoding a dominant negative version of cadherin (*dn-cadherin*) that blocks Wnt signalling by sequestering βCatenin at the level of the membrane, largely eliminated *ksr3* expression and prevented formation of the PMCs. These results are consistent with the idea that *beta catenin →pmar1−<hesC−<ksr3*. Injection into the egg of a morpholino oligonucleotide targeting the translation start site of the *ksr3* transcript caused a dose-dependent block of PMC ingression into the blastocoele and strongly reduced or suppressed activation of ERK in the PMCs (Fig. 4b). We confirmed these results using a second morpholino targeting the exon4-intron4 splice-junction of the *ksr3* transcript (Mo *ksr3* splice) (Fig. 4b, Supplementary Fig. 6). Injection of this splice site blocking morpholino produced a very consistent phenotype characterized by the almost complete absence of ERK activation at the vegetal pole of blastula stage embryos and by the absence of PMCs at the mesenchyme blastula stage. RT-PCR analysis of the structure of the *ksr3* transcript produced in the splice site-blocking morpholino injected embryos showed that the oligonucleotide caused production of an aberrantly spliced transcript generated by utilisation of a cryptic splice donor site 36 bp upstream of the normal E4I4 splice junction. Sequence analysis revealed that use of this cryptic site caused production of an mRNA that encoded a protein deleted from the entire P-loop region of KSR3 (Supplementary Fig. 6a). To test the specificity of the splice blocking morpholino we performed several experiments. First, we attempted to rescue the lack of ERK activation caused by injection of the splice site-blocking morpholino by injection of mRNA encoding an activated form of KSR3 not recognized by the morpholino (see below). While embryos injected with the morpholino alone failed to activate ERK at the vegetal pole, embryos injected with the morpholino and the mRNA showed strong activation of ERK (Supplementary Fig. 6b). We also tested if the translation blocking and the splice site-blocking morpholinos displayed synergistic effects. While injection of low doses of each morpholino did not block PMC ingression and ERK activation, co-injection of the two morpholinos at the same doses very efficiently blocked PMC ingression and suppressed ERK activation (Supplementary Fig. 6c).

### KSR3 is required for the ability of Pmar1 to convert any cell of the embryo into a PMC
We next tested if injection of morpholino oligonucleotides targeting *ksr3* blocked the effects of *pmar1* overexpression or of those caused by inhibition of *hesC*. Remarkably, while embryos overexpressing *pmar1* or embryos injected with the *hesC* morpholino started to massively extrude PMCs at the mesenchyme blastula stage as a consequence of excessive PMC production and massive EMT induced by these factors, embryos injected simultaneously with *pmar1* mRNA or *hesC* morpholino and with the morpholino targeting *ksr3* did not undergo EMT but remained as empty balls of epithelial cells (Fig. 4b). This shows that the function of KSR3 is required for Pmar1 to induce its dramatic

morphogenetic effects. These results strongly suggest that zygotic transcription and translation of *ksr3* mRNA are necessary steps for the RAS-independent activation of ERK in the PMCs downstream of the Pmar1/HesC double negative gate (Fig. 4e).

### KSR3 R364H is a potent dominant negative version of KSR3
Finally, we tested whether sea urchin KSR1 and KSR3 heterodimerize with RAF. We constructed mutant forms of sea urchin KSR1 and KSR3 in which the highly conserved arginine residue required for dimerization with RAF was replaced with histidine (generating KSR1-R608H and KSR3-R364H) and injected the mRNAs encoding these factors into sea urchin eggs[7] (Fig. 4b). In embryos injected with the KSR1-R608H mutant the PMCs formed normally and ERK was activated in the skeletogenic precursors at the vegetal pole. In contrast, embryos injected with the KSR3-R364H mutant did not form PMCs and no activation of ERK was detected at the vegetal pole. This phenotype is identical to that caused by injection of morpholino oligonucleotides targeting the *ksr3* transcript. This suggests that the R364H mutant of KSR3 is not only a loss-of-function mutant but that it has a potent dominant negative activity that blocks the function of endogenous KSR3.

### KSR3 requires dephosphorylation on the N-terminus to be active
To test whether *ksr3* is the downstream effector of Pmar1 and whether it is sufficient to activate ERK signalling, we overexpressed it by mRNA injection into fertilized eggs. Surprisingly, overexpression of wild-type *ksr3* did not cause massive ectopic activation of ERK or massive EMT (Fig. 4b). Because the activity of KSR factors is negatively regulated by the binding of 14-3-3 proteins to the phosphorylated serine at the N-terminal region (serine 259)[5,32–35], and positively regulated by dephosphorylation by PP2A, this raised the possibility that sea urchin KSR3 activity is repressed by a similar mechanism. Indeed, overexpression of a mutant form of KSR3 in which the two serine residues (Ser84, Ser87) (Fig. 2c) in position equivalent to Ser259 of human B-RAF were substituted by alanine residues caused a massive ectopic activation of ERK (Fig. 4b). Furthermore, while overexpression of *ksr3* alone or of the regulatory subunit of PP2A (*pp2aB*) alone did not cause ectopic activation of ERK, co-overexpression of *ksr3* and *pp2aB* caused a massive activation of ERK in the injected embryos Fig. 4c). In contrast, overexpression of a dominant negative version of the catalytic subunit of PP2A (pp2aC L209P) abrogated activation of ERK at the vegetal pole and inhibited formation of the PMCs confirming that dephosphorylation of KSR3 by PP2A is likely a required step in the mechanism regulating KSR3 activation. Strikingly, while overexpression of the ERK pathway activator *ksr3 S84A/S87A* alone or overexpression of the PMC specification factor *alx1* alone did not cause any visible morphological phenotype, co-injection of *ksr3 S84A/S87A* mRNA with *alx1* mRNA caused a spectacular phenotype characterized by a massive EMT and the extrusion of a large number of PMC-like cells at the vegetal pole of the embryos (Fig. 4d). This indicates that while overexpression of *alx1* alone is not sufficient to

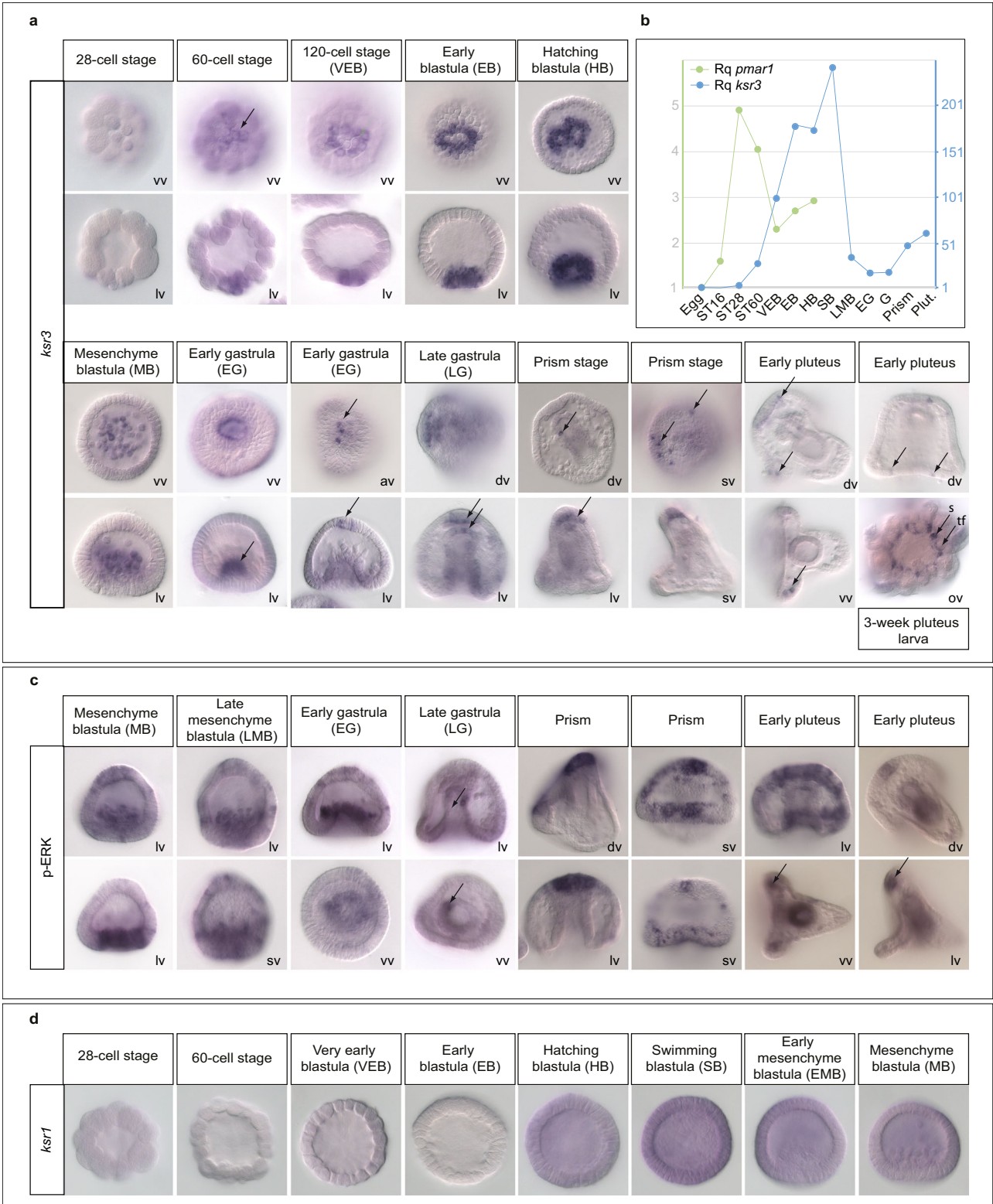

produce an excess of PMCs and activation of ERK alone is not sufficient to trigger a massive EMT, when combined, these factors both increase the number of PMCs specified and trigger their delamination resulting in the massive EMT phenotype.

### *ksr3* is the founding member of an evolutionary conserved family of atypical *ksr* genes present in non-chordate metazoa

So far, the finding of an additional structurally unique KSR3 factor expressed in the sea urchin skeletogenic mesoderm as well as its

unusual role in the cell autonomous activation of ERK signalling suggested that *ksr3* was likely a gene produced by a recent duplication of a *ksr* gene followed by a functional specialization in skeletogenesis. Such a mechanism has been proposed for example in the case of the *alx1/ alx4* genes and for the *pmar1* gene, which is only found in the lineage of echinoids, in which it has evolved by gene amplification[36,37]. We therefore expected *ksr3* to be a gene present only in sea urchin species that build an embryonic skeleton from micromeres and PMCs and that possess the double negative gate circuit. To test this hypothesis, clarify

**Fig. 3 | *ksr3* is expressed dynamically during blastula, gastrula and pluteus stages, first in the skeletogenic mesoderm, then during gastrulation in the endoderm, animal pole-and ciliary band, then in the adult rudiment at pluteus stage and its expression overlaps with domains of ERK activation. a** Time course of *ksr3* expression during early development and in the adult rudiment. In situ hybridizations with a *ksr3* probe during blastula, gastrula, prism and pluteus stages. *ksr3* is first expressed transiently in the PMCs. Note that at 64-cell-stage and very early blastula stages there are 8 micromeres expressing *ksr3* while at early blastula and swimming blastula stages there are respectively 16 and 32 PMC precursor cells expressing this gene. During gastrulation, *ksr3* is expressed in the invaginating archenteron, in neural cells, most likely serotonergic neurons, of the

animal pole. In 3-week old pluteus larvae, *ksr3* is expressed in the developing appendages, spines (s) and tube feet (tf) of the adult rudiment. lv lateral view, dv dorsal view, vv vegetal view, ov oral view, sv surface view. Arrows point to the domains of *ksr3* expression. **b** QPCR analysis of *ksr3 and pmar1* expression. *ksr3* is expressed transiently in the PMC precursors immediately after *pmar1*. **c** phospho-ERK immunostaining during gastrulation. The territories in which *ksr3* is expressed are largely overlapping with domains of ERK activation. Arrows point to the domains of ERK activation. **d** *ksr1* is expressed at a low level and ubiquitously in different germ layers during development. The number of times the experiments were replicated and the number of embryos analyzed are indicated in See Supplementary Table 1.

the relationships between KSR3 proteins and the other KSRs and to shed light on the emergence of vertebrate *ksr1* and *ksr2* genes in the course of evolution, we catalogued *ksr* and *raf*-related genes from 31 species representing major metazoan clades and reconstructed a phylogenetic tree (Fig. 5a, b). This analysis revealed that at least one copy of *ksr1* and one copy of *raf* are present in the genome of all metazoans. This analysis also revealed that the two *ksr* genes from vertebrates, *ksr1* and *ksr2*, are paralogs produced by a recent duplication of the *ksr1* gene. The same is true for Nematodes, which appear to possess two recently duplicated *ksr1* genes. This phylogenetic analysis also revealed that in addition to canonical *raf* and *ksr1* sequences, the genomes of most non-chordate metazoa contain an additional *ksr* gene that is not a paralog of *ksr1* or *ksr2* but that is structurally highly similar to the *ksr3* gene that we isolated (Supplementary Fig. 1, see Supplementary data set 1 for a full alignment of RAF and KSRs sequences). Orthologs of *ksr3* gene could be detected in most major animal clades including in the diploblastic Cnidaria (*Acropora millepora*, *Nematostella vectensis* and *Clytia hemispherica*) that branch basally in the tree of life and in Placozoa (*Trichoplax adherens*). Within Protostomia, a gene encoding a KSR3 factor could be identified in the genome of Arachnids (*Rhipicephalus sanguineus*), of Crustaceans (*Limulus Polyphemus*) and of Priapulids (*Priapulus caudatus*). *ksr3* genes were also detected in flatworms (Platyhelminths (*Schmidtea mediterranea*)) and in representative of Spiralia such as Annelids (*Capitella teleta*) and Molluscs (*Crassostrea gigas*). Finally, *ksr3* was present in the genomes of all non-chordate deuterostomes we searched, including Echinoderms from different classes such as sea urchins (*Paracentrotus lividus*, *Stronglyo-centrotus purpuratus*, *Eucidaris tribuloides*), starfish (*Patiria miniata*), crinoids (*Anessia japonica*), sea cucumbers (*Parastichopus parvi-mensis*), ophiurids (*Amphiura filiformis*) as well as in Hemichordates (*Saccoglossus kowaleskii*, *Ptychodera flava*). In contrast, *ksr3* appears to be absent from the genomes of Sponges (*Amphimedon queenslandica*) as well as from *C. elegans* and *D. melanogaster*, two model organisms that are known to have lost a number of genes[38]. Similarly, we could not retrieve *ksr3* sequences from Cephalochordates, Tunicates and Vertebrates genomes and transcriptomes. The data are therefore consistent with the idea that *ksr1* and the additional *ksr3* gene that we have characterized are ancient genes that may have appeared by gene duplication in Eumetazoa, soon after the emergence of multicellular organisms. Although both *ksr1* and *ksr3* have been maintained in the genomes of protostomes and non-chordate deuterostomes, *ksr3* was subsequently lost in insects and nematodes and, more recently, it was also lost in the course of the evolution of chordates.

## KSR3 proteins define a previously uncharacterized family of B-RAF activators

The predicted structures of all the KSR3 factors identified differed markedly from that of previously described KSR proteins from *Drosophila* or vertebrates (Fig. 2c and Supplementary Data 1). Not only KSR3 proteins lacked the proline-rich domain and the N-terminal coiled-coil sterile alpha motif (CC-SAM domain) that is present in KSR1 proteins and that is thought to bind to the N-terminal region of B-RAF factors[28], but the sequence of KSR3 proteins also differed significantly

from those of prototypical KSR or from B-RAF proteins at the level of the kinase domain. Comparison of the sequence of the kinase domain of KSR3 factors with that of B-RAF revealed that the most variable positions of KSR3 clustered in discrete regions: at the level of the P-loop, the alpha-C helix and the activation segment, i.e. the same regions as those where the gain-of-function mutations of B-RAF have been mapped (Fig. 5c, d)[39,40]. For example, alignment of the KSR3 sequences revealed that structurally, KSR3 factors are naturally occurring variants of the G464E/V/R and G469A/V/S P-loop oncogenic mutations of B-RAF. Furthermore, when compared to B-RAF or KSR1/2 factors, the sequences of most KSR3 proteins diverge in the P-loop region at positions equivalent to the conserved F468 mutations of B-RAF (F468K/R/N/Q/D), mutations that are detected in a number of cancers[39,40]. Most KSR3 proteins also appeared to carry mutations in the alpha C helix that replaced highly conserved leucine residues (L485 from B-RAF) by arginine (R343 of sea urchin KSR3) or highly conserved lysine (K499 of B-RAF) by leucine (L354 of sea urchin KSR3). All KSR3 factors also lacked the prototypical "DFG" motif of the activation loop, which is essential for the autoinhibition mechanism since it interacts with the P-loop and blocks the ATP binding site in the inactive state, and most of them carried variable residues in positions equivalent to D594 from human B-RAF (such as the D594G mutation, which is oncogenic in humans) or to L597 (such as L597H,T,G,N,M,F,V,T) of the activation segment of B-RAF, which are also mutations found in a number of cancers[39–41]. Finally, KSR3 factors lacked the ERK phosphorylation sites that are involved in the negative feedback regulation of the activity of B-RAF and KSR1 (Supplementary Fig. 5)[42,43]. Taken together, these observations suggested that the structure of KSR3 factors resembles that of oncogenic or of overactivated B-RAF mutants, raising the possibility that KSR3 may be natural constitutively active activators of RAF and ERK signalling.

## Overexpression of KSR3 family members is sufficient to activate ERK signalling in a RAS-independent manner in cultured human cells

To test if KSR3 factors can activate ERK signalling when overexpressed in cultured cells, we overexpressed KSR3 factors from three different species in HEK293T human cells and monitored ERK activation (Fig. 6). Indeed, overexpression of sea urchin *ksr3* alone, but not sea urchin *ksr1*, could drive a robust activation of ERK signalling that was similar in intensity to that caused by overexpression of an activated RAS (G12V) mutant or to the Delta NVTAP B-RAF class II mutant[44] (Fig. 6a, b). Similarly, overexpression of *ksr3* from Hemichordates (*Saccoglossus kowaleskii*) or Cnidarians (*Nematostella vectensis*), was also sufficient to drive strong activation of ERK in cell culture (Fig. 6c). Consistent with the absence of a RAS binding domain and of a CC-SAM domain in KSR3 factors, these effects were largely independent of activation of RAS and were not inhibited by overexpression of a dominant negative form of RAS (Fig. 6d and Supplementary Fig. 7e). In contrast, the strong activation of ERK caused by overexpression of *ksr3* from sea urchin, Hemichordate or Cnidarian, was critically dependent on RAF since it was largely suppressed by overexpression of a dominant negative form of human C-RAF (Fig. 6d). We confirmed that the ability

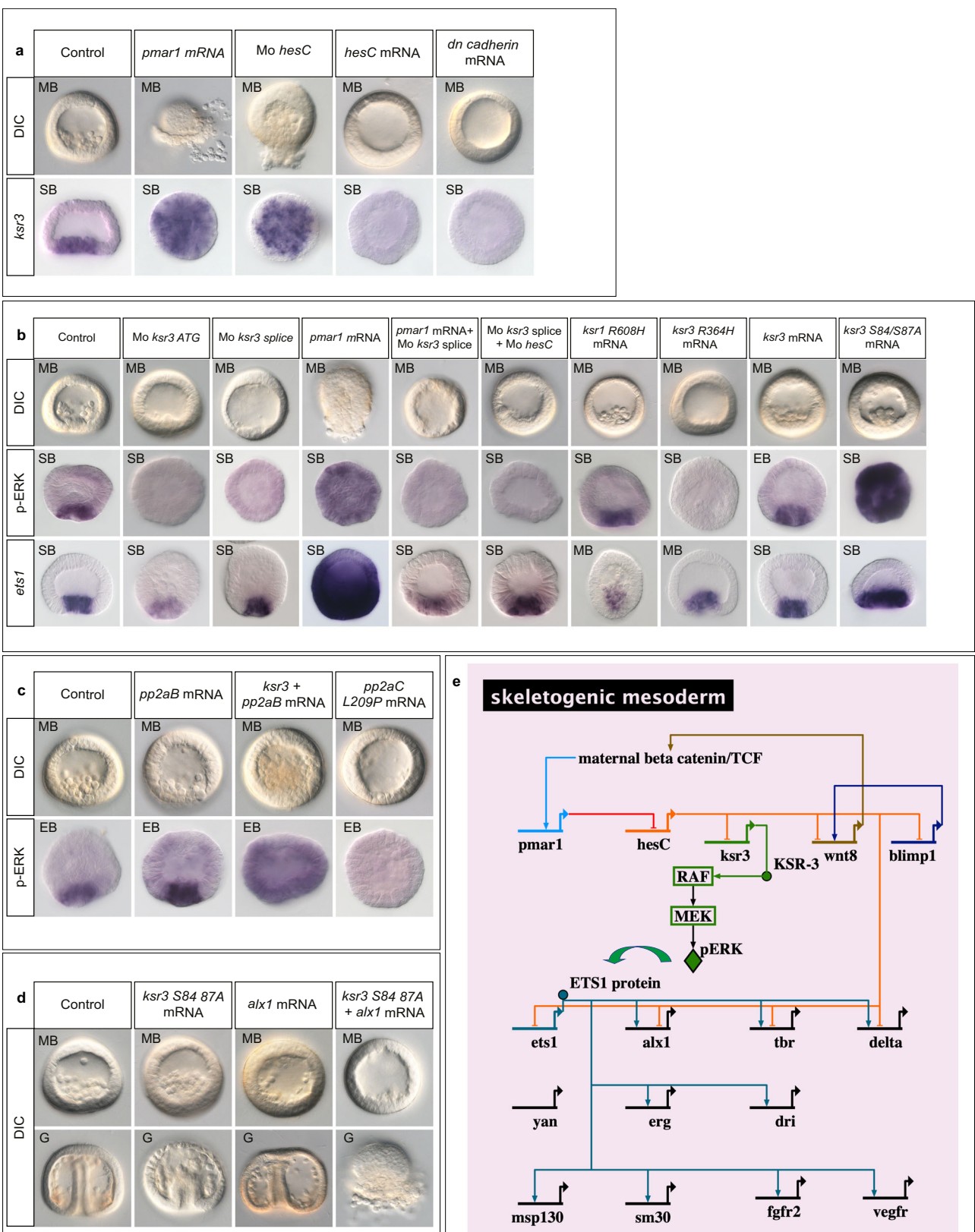

of KSR3 factors to activate ERK signalling requires dimerization with RAF. We constructed and overexpressed mutant forms of sea urchin, hemichordate and cnidarian KSR3 in which the highly conserved arginine residue required for dimerization with RAF was replaced with histidine. For all three KSR3, mutation of the highly conserved arginine abolished their ability to activate ERK signalling (Fig. 6e). Finally, co-immunoprecipitations experiments with sea urchin KSR3 and RAF factors indicated that KSR3 can form complexes with RAF (Fig. 6f). These observations identify KSR3 family members as a family of evo-lutionary conserved RAS-independent activators of RAF.

**Fig. 4 | KSR3 is required for activation of ERK in the PMCs and for the ability of Pmar1 to convert any cell of the embryo into PMCs. a** Expression of *ksr3* is positively regulated by Pmar1 and repressed by HesC. Injection of *pmar1* or injection of an antisense morpholino oligonucleotide targeting the *hesC* transcript caused a massive ectopic expression of *ksr3* while overexpression of *hesC* mRNA abolished *ksr3* expression. **b** Injection of antisense morpholino oligonucleotides targeting the *ksr3* translation start site (*Mo-ksr3ATG*) or targeting the exon4-intron4 junction (*Mo-ksr3 splice*) prevented activation of ERK in the skeletogenic precursors. While overexpression of *pmar1* induced a massive activation of ERK, co-injection of *pmar1* mRNA and of *ksr3 splice* morpholino suppressed the *pmar1*-induced activation of ERK, resulting in embryos devoid of PMCs. Injection of the *ksr3 splice* morpholino also suppressed the massive EMT and activation of ERK caused by injection of a morpholino oligonucleotide targeting the HesC transcript. Injection of mRNA encoding a dimerization defective KSR1 factor (KSR1 R608H) did not block activation of ERK in the PMC precursors and did not perturb EMT of these cells. In contrast, injection of mRNA encoding a dimerization defective KSR3 factor (KSR3-R364H) caused the same phenotype as that caused by the *ksr3* morpholinos,

blocking activation of ERK and PMC delamination. Overexpression of *ksr3* does not cause ectopic activation of ERK. However, substitution of alanine for serine 84 and serine 87 of KSR3 produced a constitutively active KSR3 factor that was able to ectopically activate ERK signalling in the embryo. EB early blastula, SB swimming blastula, MB mesenchyme blastula, G Gastrula, DIC Differential Interference Contrast. **c** Co-overexpression of *ksr3* and of the regulatory subunit of PP2A induces massive activation of ERK while overexpression of a dominant negative form of the catalytic subunit of PP2A abrogates activation of ERK and EMT. **d** While single overexpression of the ERK pathway activator *ksr3* or of the PMC specification gene *alx1* do not cause any morphological phenotype, co-overexpression of *alx1* and *ksr3* induces a massive production of PMCs and a large wave of EMT at the vegetal pole. **e** Biotapestry[82] diagram showing position of the genetic circuit responsible for activation of ERK signalling in the PMC gene regulatory network: *ksr3* is a target of the Pmar1-HesC double negative gate. The number of times the experiments were replicated and the number of embryos analyzed are indicated in See Supplementary Table 1.

## KSR3 requires phosphorylation of the acidic N-terminal motif (NtA) and is regulated at the C-terminus by 14-3-3 proteins

Activation of RAF factors requires phosphorylation of two regions: an N-terminal region immediately upstream of the kinase domain referred to as the acidic N-terminal region (NtA) and the activation loop (AL) or activation segment (AS)[45–47]. To further test whether the NtA region of KSR3 proteins is required for activation of ERK signalling by these factors, we substituted alanine residues for the acidic and serine residues present in the NtA motifs of sea urchin and *Saccoglossus* KSR3 creating respectively SE > AA and DE > AA mutants. Mutation of the NtA motifs of these two KSR3 factors abrogated their ability to activate ERK signalling revealing that these residues are crucial for the ability of KSR3 to activate ERK (Fig. 6h and Supplementary Fig. 7c).

In addition to the NtA region, phosphorylation of the activation loop is required for activation of RAF[48–50]. KSR3 factors contain acidic residues in position equivalent to T491 of the activation segment of human B-RAF (Supplementary Fig. 3) raising the possibility that they may contribute to the constitutive activity of KSR3 by functioning like the V600E mutation of human b-RAF. Substitution of the acidic residues of the activation loop with alanine generating the AS D452, D455A double mutant or substitution of the four serine and threonine residues present in the activation segment with alanine (AS (AAAA)) did not reduce the ability of KSR3 to activate ERK signalling (Supplementary Fig. 7c). Taken together these results suggest that the ability of sea urchin KSR3 to activate ERK signalling is critically dependent on the presence of a negatively charged NtA region but is probably largely independent of phosphorylation of the activation loop or of the presence of acidic residues in this region. Finally, RAF and KSR1 factors also critically require phosphorylation of the C-terminal region followed by recruitment of 14-3-3 proteins for dimerization and transactivation[32,34,35,51,52]. Mutation of the potential 14-3-3 binding site in the sea urchin, hemichordate or cnidarian KSR3 factors largely abolished their ability to activate ERK signalling (Fig. 6g). This suggests that the positive regulation of KSR3 factors by 14-3-3 proteins binding to the C-terminal region is a very ancient feature that has been conserved since the divergence of cnidarians from the ancestor of bilaterians.

## The lack of a CC-SAM domain, a shorter alpha C helix and a set of KSR3 specific amino acids are responsible for the constitutive activity of KSR3

We next investigated what are the structural determinants that confer to KSR3 their ability to constitutively activate ERK signalling. The mechanism leading to activation of KSR or RAF involves release of the autoinhibitory action of the N-terminal region on the kinase domain[5]. KSR3 lacks the N-terminal CC-SAM domain, suggesting that the absence of the CC-SAM domain may contribute to its activity. Addition of the SAM domain of sea urchin KSR1 to the sea urchin KSR3 protein

slightly decreased its ability to activate ERK (Supplementary Fig. 7 construct *Pl* SAM KSR1 + KSR3) consistent with the idea that the N-terminal domain of KSR proteins inhibits the kinase domain. We also noted that most KSR3 proteins have a β3 alpha C region 3 residues shorter than B-RAF or KSR1 factors (Fig. 7a). In frame deletions of the β3 alpha C region of B-RAF have been shown to activate A-RAF, B-RAF and C-RAF by constraining the alpha C helix in an active "In "conformation[44,53]. To test if the shorter length of the β3 alpha C region of KSR3 contributes to its constitutive activity, we inserted 3 amino acids in its β3 alpha C region to make it of the same length as the β3 alpha C region of human B-RAF (construct *Pl* KSR3 VTA[insertion 345]) (Fig. 7b). As expected lengthening the β3-alpha C region decreased by about 50% the ability of KSR3 to activate ERK in transfected HEK293T cells. Finally, we mutated most of the residues that are specific to KSR3 within the β3, alpha C, DIF and β7- β8 regions (Fig. 7a, b). In B-RAF, L485, on strand β3, is part of hydrophobic network that comprises F497 on the helix alpha C, L597 and V600 on the AS[9]. Intriguingly, most KSR3 factors have instead an arginine (R343 in sea urchin KSR3) in the position homologous to L485 in human B-RAF. Substitution of R343 of sea urchin KSR3 by L strongly reduced the ability of KSR3 to activate ERK signalling (mutant R343L in Fig. 7b). Similarly, within the alpha C helix, most RAF proteins have a lysine in position 499 while KSR3 factors have a leucine or a tryptophan residue. Substitution of L354 of sea urchin KSR3 into K decreased the ability of KSR3 to activate ERK to 25% its normal value (mutant KSR3 L354K in Fig. 7b). Finally, within the β7- β8 region, replacing lysine 443 by leucine or valine 444 by glycine similarly impaired the ability of sea urchin KSR3 to activate ERK (mutants Pl KSR3 K443L and Pl KSR3 V444G in Fig. 7b). Similarly, within the dimerization interface of KSR3, mutations K362E and I363T severely reduced the ability of sea urchin KSR3 to activate ERK but unexpectedly, replacing S361 of sea urchin KSR3 by a glutamic acid had the opposite effect: it increased the ability of KSR3 to activate ERK signalling about two-fold and this effect was abolished in the presence of the R364H mutation (Fig. 7b, c, mutant *Pl* KSR3 S361E and *Pl* KSR3 S361E R364H). Interestingly, the S361E mutation of KSR3 partially restored the ability of a KSR3 NtA mutant to activate ERK signalling, strongly suggesting that the S361E mutation bypasses the requirement for a phosphorylated NtA region by triggering transactivation of B-RAF (Fig. 7c mutant *Pl* KSR3 S361E SE > AA).

## KSR3 function requires an intact CRD

Finally, since KSR3 contains a N-terminal cysteine rich domain (C1 or CRD) and since CRD domains have been implicated in the regulation of membrane localization of KSR proteins by binding to phospholipids[54], we mutated the conserved cysteines of KSR3 and tested the effect of these mutations on its ability to activate ERK signalling. Replacing cysteines 37 and 40 of KSR3 by serine strongly reduced the ability of

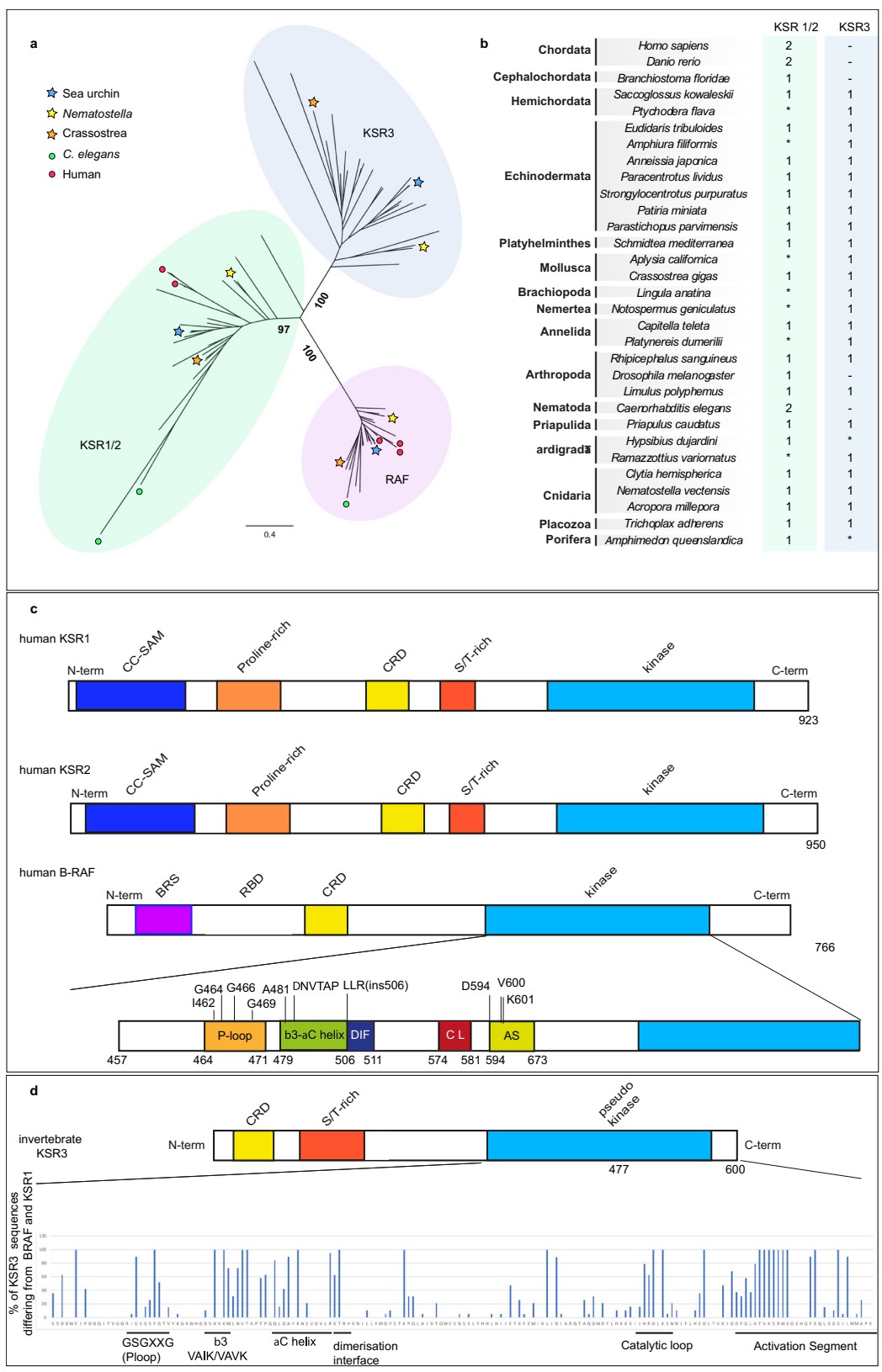

KSR3 to activate ERK (Fig. 7d construct *Pl* KSR3 C37/40 S). Taken together these results demonstrate that most of the residues that are present in KSR3 but absent in B-RAF or KSR1 are in fact essential for its ability to activate ERK signalling suggesting that they may have been selected in the course of evolution to confer to KSR3 the properties of a constitutive activator of ERK signalling.

**Replacing amino acids of B-RAF within the β3, alpha C, DIF and β7-β8 regions by amino acids found at homologous positions in KSR3 strongly activates human B-RAF**

Starting from the idea that KSR3 proteins have evolved during hundreds of millions years to become constitutive activators of ERK and since KSRs and B-RAF share similar structures, we reasoned that

**Fig. 5 | *ksr3* is an ancient gene present in most non-chordate phyla and the structure of KSR3 factors resembles that of oncogenic human RAF mutants.** **a** Maximum likelihood phylogenetic analysis of KSR and RAF sequences showing that *ksr3* sequences constitute an ancient family of genes conserved in metazoans. Numbers represent bootstrap values calculated from a consensus of 1000 replicates[77, 78] (see Supplementary data Fig. 1 for a detailed tree). **b** *ksr1/2* and *ksr3* gene complement across metazoans. Numbers indicate the presence and number of *ksr1/2* and *ksr3* genes. Asterisks indicate that a complete *ksr* complement could not be confidently determined. Note that *ksr3* has been lost in chordates as well as in nematodes and insects. Also note that the nematode and human *ksr1* and *ksr2*

genes are in fact recent duplications of an ancestral *ksr1* gene. **c** Schematic representation of the structure of human KSR1, human KSR2, human B-RAF and of KSR3 protein from sea urchin. In humans, most oncogenic mutations map to specific regions of B-RAF such as the P-loop, the alpha C helix, the dimerization interface and the activation segment. **d** Frequency of positions in KSR3 that differ from B-RAF within the set of 19 KSR3 sequences selected for the analysis. KSR factors mainly diverge from B-RAF at the level of the P-loop, the alpha C helix, the dimerization interface, the catalytic loop and the activation segment. The sequence of human B-RAF is shown as a reference. The histogram shows the percentage of divergence compared to B-RAF.

residues that are uniquely found in key regions of KSR3 factors might activate B-RAF as well when swapped at homologous positions (Figs. 7a, 8a, d). We first tested this idea by replacing L485 of human B-RAF by amino acids found in various KSR3 at this position (F, Y or R). Indeed, replacing L485 by F increased significantly the activity of B-RAF as reported previously[9] (Fig. 8a mutant *h*B-RAF L485F). Strikingly, replacing L485 of B-RAF by R, as in echinoderm KSR3, increased even more dramatically the activity of B-RAF, which reached a level similar to that of the V600E mutant (Fig. 8a mutant *h*B-RAF L485F). Mutations of L485 such as L485F/W/S have been found in several cancers (COSMIC database https://cancer.sanger.ac.uk/cosmic) but L485R has never been described so far, most likely because it is a rare mutation that requires two substitutions. Similarly, deletion of Valine 487 of B-RAF (mutant *h*B-RAFΔV487), substitution of the QLQA motif of the alpha C helix by the more acidic DIND sequence of KSR3 (mutant *h*-BRAF QLQA > DIND) modestly but reproducibly increased the activity of B-RAF. In contrast, replacing K499 of the alpha C helix of B-RAF by a leucine (mutant *h*B-RAF K499L), as in KSR3, as well as the R506S, D587K, L588V mutations all caused strong activation of human B-RAF (Fig. 8a mutants *h*B-RAF R506S, hB-RAFD587K, *h*B-RAF L588V). Again, we note that mutations of K499, R506 and L588 have been found in several cancers (data from COSMIC database). The highest level of activation of ERK signalling was observed by transposing on B-RAF the S361E activating mutation of KSR3. Like in the case of the L485R mutant, the resulting R506E mutant resulted in a form of B-RAF that had a level of activity comparable to that of the V600E mutant (Fig. 8a mutant *h*B-RAF R506E). As with B-RAF L485R, the R506E mutation of B-RAF has never been described possibly because it is rare and requires two substitutions. We tested the effect of the central R506H mutation on these B-RAF mutants and found that, unlike the KSR3 S361E mutant, which was fully sensitive to the R364H mutation, all of them retained a considerable level of activity in the presence of the R509H mutation, with in the decreasing order of resistance V600E ≥ L485R ≈ R506E > K499L > R506S > WT (Fig. 8b mutants *h*B-RAF L485R R509H, *h*B-RAF K499L R509H, *h*B-RAF R506S R509H, *h*B-RAF R506E R509H). Finally, as in the case of the KSR3 S361E, the B-RAF R506E restored significantly the ability of a B-RAF NtA mutant (mutant *h*B-RAF R506E 4 A) to activate ERK signalling strongly suggesting that the KSR3 S361E and B-RAF R506E mutations are activating KSR3 and B-RAF through identical mechanisms, possibly by strengthening their ability to transactivate B-RAF protomers (Fig. 8c).

**KSR3 S361E and B-RAF R506E display increased dimerization and increased transactivation potential**

Dimerization and transactivation of RAF kinases and KSR factors is critical for their activity. We, therefore, investigated whether the three strongest mutants that we found, KSR3 S361E, B-RAF L485R and B-RAF R506E, activate KSR3 or B-RAF by promoting dimerization or by enforcing transactivation. To test if these mutations promote dimerization, we used a co-immunoprecipitation assay. Wild type sea urchin 3HA-KSR3 efficiently co-immunoprecipitated WT sea urchin MYC-tagged B-RAF (Fig. 9a). The KSR3 S361E mutant increased about 1.5-fold the amount of WT B-RAF co-immunoprecipitated in this assay. Similarly, wild type human HA-B-RAF

efficiently co-immunoprecipitated human MYC-tagged B-RAF and the R506E mutant of B-RAF increased 1.3-fold the amount of B-RAF co-immunoprecipitated indicating that these two mutations increase the dimerization potential of KSR3 and B-RAF (Fig. 9b mutant *h*-BRAF R506E +*h*B-RAF). In contrast, *h*B-RAF L485R apparently did not increase the amount of B-RAF immunoprecipitated (mutant *h*B-RAF L485R +hB-RAF. We then tested whether these KSR3 and B-RAF mutants promote instead transactivation of B-RAF protomers in a transactivation assay (Fig. 9c, d).

When sea urchin KSR3 (kinase-defective activator) was co-transfected with a human B-RAF AAAA mutant receiver (kinase-active), it activated ERK signalling by the receiver consistent with the idea that KSR3 is a strong activator of B-RAF. Introduction of the S361E mutation in KSR3 resulted in a sligthly stronger stimulation of the receiver suggesting that it is indeed a slightly more potent transactivator of B-RAF than KSR3 WT (Fig. 9c lane 5 mutant *Pl* KSR3 S361E +h-B-RAF4A)).

Similarly, while a kinase-defective form of human B-RAF (B-RAF A481F) very modestly activated ERK signalling when co-transfected with a B-RAF receiver (B-RAF AAAA), the same kinase-defective B-RAF mutant strongly activated ERK signalling through the B-RAF receiver when combined to the R506E mutation (Fig. 9d lane 6 mutant *h*B-RAF A481F R506E +*h*B-RAF4A). These results demonstrate that KSR3 S361E and B-RAF R506E significantly bypass the requirement for a phosphorylated NtA region for activation in "trans" of B-RAF, possibly by mimicking the effects of the phosphorylated NtA motif and possibly by directly triggering efficient allosteric transactivation of the B-RAF protomers.

## Discussion

**Identification of a missing link in the gene regulatory network controlling endomesoderm formation in the sea urchin embryo**

In recent years our knowledge of the GRN that regulates specification of the primary mesenchymal cell precursors and epithelial-mesenchymal transition has increased significantly. Screens have identified genes specifically expressed in these cells[55], genes required for their delamination, and transcription factors required for their movements[56]. Genome wide screens also identified targets of key transcription factors and several genes that require MAPK signalling for their expression[57,58]. However, these approaches did not identify the gene(s) downstream of beta Catenin that triggers activation of MAPK signalling, which, in the PMCs precursors of the early sea urchin embryo, does not appear to require cell interactions and is independent of the activity of RAS. Few examples of RAS-independent activation of MAPK signalling have been described and in the vast majority of cases, ERK signalling is activated by RTK following binding of growth factors or following a cellular stress. The identification of *ksr3* as the gene that is activated by Pmar1 and that is responsible for activation of ERK signalling fills this gap in our knowledge the PMC GRN. In addition, the identification of *ksr3* as a gene whose expression is sufficient to turn on the ERK pathway raises interesting questions related to the evolution of this GRN in echinoderms and suggests a possible role for this gene in the emergence of biological novelties and in the co-option of the ERK pathway in the course of evolution.

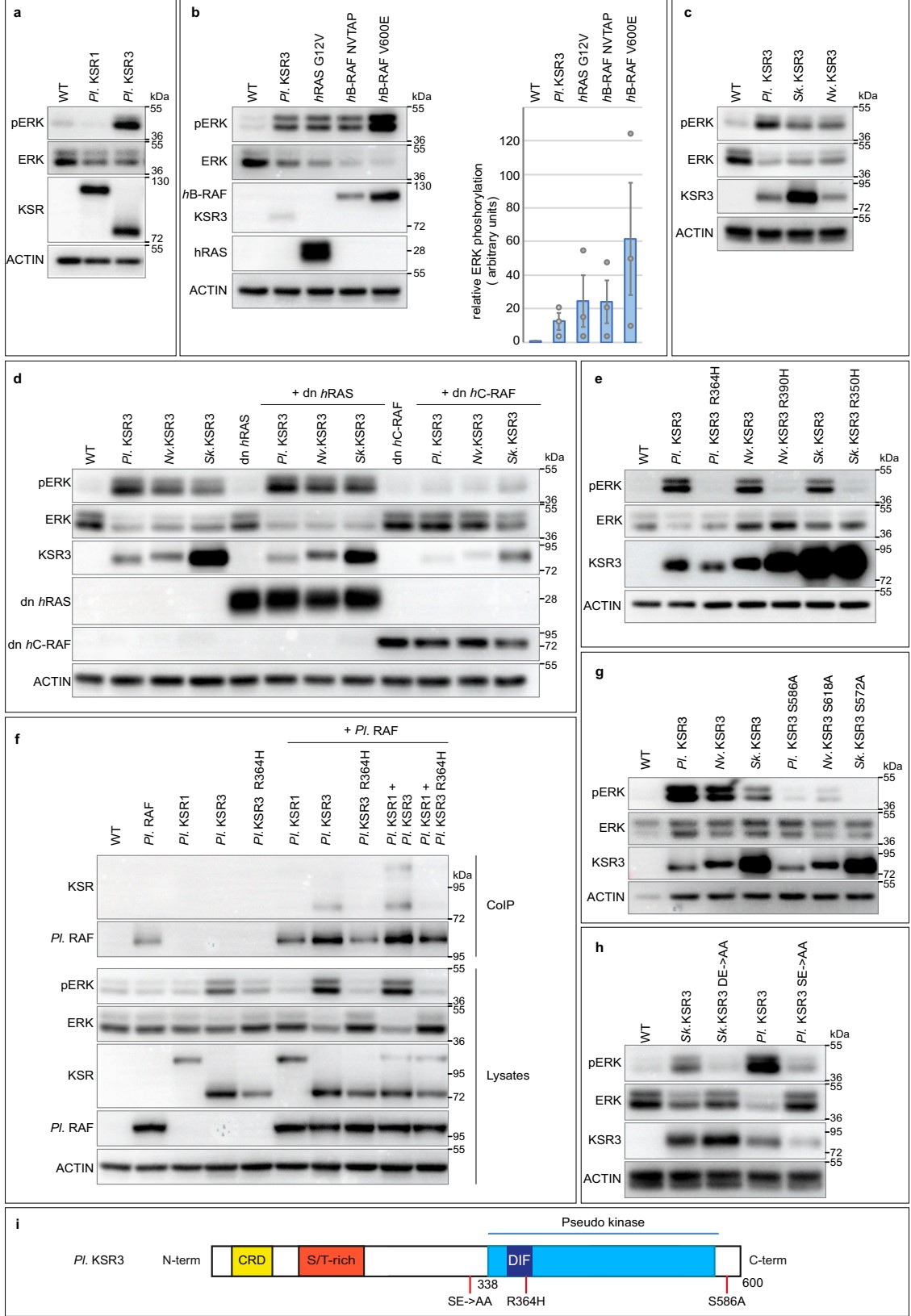

### KSR3 as the signal that activates ERK signalling in the sea urchin embryo

Previous studies have shown that activation of ERK required the activity of TCF/ βCatenin, did not require cell interactions, was RAS-independent and nevertheless required the activity of RAF and MEK[12]. In this paper, we provide several lines of evidence that *ksr3* is the gene responsible for the RAS-independent activation of ERK signalling downstream of TCF/ catenin. First, we have shown that *ksr3* expression closely follows activation of the β Catenin target gene *pmar1* in the PMC lineage and that Pmar1 acts upstream and regulates the expression of *ksr3*. Moreover, we have shown that *ksr3* expression is very precocious and precedes activation of ERK signalling in the progeny of

**Fig. 6 | KSR3 factors act as RAS-independent constitutively active triggers of ERK signaling. a** Overexpression of sea urchin KSR3 (Pl. KSR3) but not of KSR1 (Pl. KSR1) triggers ERK signalling in Human HEK293T cells. **b** Overexpression of sea urchin KSR3 activates ERK signalling at the same level as overexpression of activated RAS G12V or of an activated B-RAF deletion mutant (Delta NVTAP). **c** KSR3 from *Saccoglossus kovaleskii* (Sk. KSR3) and from *Nematostella vectensis* (Nv. KSR3) also activate ERK signalling when overexpressed. **d** KSR3 from sea urchin, hemichordate and cnidarian activate ERK signalling in a RAS-independent but RAF-dependent manner. **e** KSR3 from echinoderm, hemichordate and cnidarian require dimerization to activate ERK signalling. Substitution of a highly conserved arginine residue required for dimerization by histidine suppressed the ability of KSR3 proteins to activate ERK signalling. **f** Sea urchin KSR3 (Pl KSR3) interacts with sea urchin RAF in co-immunoprecipitation experiments. **g** KSR3 factors from *Paracentrotus*, *Saccoglossus* and *Nematostella* require interaction with 14-3-3 proteins at the C-terminus. Substitution of alanine for a serine residue located in a potential 14-3-3 binding site of sea urchin (sequence of the motif: HSLSEP), hemichordate (HSQ**S**EP) or cnidarian (GSHSEP) strongly impairs the ability of these factors to activate ERK signalling. **h** Phosphorylation of the NtA motif of KSR3 factors is essential for their ability to activate ERK signalling. Substitution of alanine for the aspartate and glutamate residues 293, 294 of *Saccoglossus* KSR3 and for serine 306 and glutamate 307 of *Paracentrotus* KSR3 strongly impairs their ability to activate ERK signalling. **i** Scheme describing the different mutations analysed. Source data are provided as a Source Data file. Information about quantification and replication of the western blots are provided in Supplementary Table 1.

the large micromeres. KSR3 was therefore an excellent candidate for a factor acting upstream of ERK activation and downstream of Pmar1 based on its timing of expression and of its spatial pattern of expression. Second, we have shown that *ksr3* is required for activation of ERK signalling in the PMC precursors and that overexpression of a non-phosphorylatable form of KSR3 or co-overexpression of *ksr3* with PP2A induces a massive activation of ERK. Finally, overexpression of wild type *ksr3* in cell culture strongly activated ERK signalling independently of RAS. Therefore, KSR3 action is cell-autonomous and independent of Ras and thus fulfills two criteria for the activator of the ERK pathway in the PMCs. Finally, we have shown that activation of ERK signalling by KSR3 relies on the activation of RAF since mutation of a single residue required for dimerization of KSR with RAF abrogates the ability of KSR3 to activate ERK signalling. It is important to note however that overexpression of *ksr3* alone is not sufficient to trigger a massive EMT or to convert all cells of the embryo into PMCs. Similarly, overexpression of the specification gene *alx1* is not sufficient to convert all cells of the blastula into migrating PMCs. However, when co-expressed with *alx1*, *ksr3* is sufficient to transform all cells of the embryo into PMCs that undergo a massive EMT at the beginning of gastrulation. Therefore, specification and EMT of the PMCs requires both the activity of the specification gene *alx1* and the activation of the ERK pathway by KSR3.

## KSR3, the evolutionary history of specification of the skeletogenic mesoderm and the double negative gate

Although modern sea urchins (euechinoids) use a micromere-derived skeletogenic mesenchyme to build their skeleton, this is not the case of all echinoderms. Other classes of echinoderms like the ophiuroids and the ancestral sea urchins called cidaroids rely instead on a population of mesenchymal cells that delaminate from the tip of the archenteron during gastrulation[59–61]. Evolutionary studies of skeletogenesis in different classes of echinoderms further reinforced the view that micromeres and the process of activation of the skeletogenic gene regulatory network in this lineage by the double negative gate are in fact recent innovations in the evolutionary history of echinoderms[14]. The double negative gate of "modern" sea urchins required both the "invention" of micromeres and a rewiring of the skeletogenic regulatory network that placed the key genes of the network such as *alx1* and *ets1* under the repressive control of the ubiquitous repressor HesC and the gene encoding the HesC repressor itself under the control of a transcriptional repressor expressed in the micromere lineage (i.e. *pmar1*). If this biological novelty required the rewiring of the GRN that allowed key transcription factors to be expressed in the micromere lineage downstream of βCatenin, it also required the ERK pathway to be activated early and exclusively in the PMC precursors. It is striking that in modern sea urchins, co-option of the ERK pathway in the PMC precursors was accomplished by a relatively simple operation: the placement of *ksr3* under the control of the double repression circuit involving Pmar1 and HesC.

The co-option of signalling pathways during evolution of animals has been correlated with morphological novelties[62]. However, the

recruitment of a whole signalling pathway, with ligand, receptor, co-receptor and signal transduction genes to a new tissue may require a large number of regulatory mutations in the promoters and enhancers of the genes encoding the different components of the pathway and significant changes in the regulatory linkages between the signalling components. In the case of the co-option of the ERK pathway at the vegetal pole of the sea urchin embryo, it is tempting to speculate that the recruitment of a RAS-independent and cell-autonomous circuit based on *ksr3* and its placement under the control of the double negative gate was particularly well adapted for the success of this innovation because it was more parsimonious, in term of regulatory mutations, than the recruitment of a RAS-dependent mechanism that would have required spatial and temporal changes in gene expression of an RTK ligand and of its cognate RTK receptor in the PMC precursors. The recruitment of a RAS-independent and cell-autonomous mechanism of ERK activation in a new tissue by expression of *ksr3* may be a genetic mechanism used for the co-option of the ERK/MAPK signalling pathway in novel locations. In the course of evolution, the recruitment of a gene such as *ksr3* and its placement under the control of a tissue specific transcription factor may have been used repeatedly in different species to initiate the spatially restricted activation of ERK in a novel tissue followed or not by the recruitment of additional components of the RAS-dependent pathway. Therefore, this process may have facilitated the emergence of evolutionary innovations involving the activation of the MAP kinase pathway in a novel location or in a novel cell lineage[63]. However, validation of this hypothesis will require additional examples of the implication of *ksr3* in co-option of the ERK pathway and in the emergence of evolutionary innovations to be documented.

## Novel territories of ERK signalling predicted from *ksr3* expression

Since *ksr3* has the ability to activate ERK signalling independently of growth factors, the spatial-temporal expression pattern of *ksr3* may account for instances in which the spatial activation of ERK cannot be explained by the expression of RTK ligands. In cnidarians, the signals that activate ERK signalling in the planula larva are not known. ERK signalling is activated at the oral pole while ligands and receptors of the FGF pathway are expressed at the opposite aboral pole[64,65]. It will therefore be important to examine if *ksr3* is expressed in the oral region in cnidarians and whether it activates ERK signalling independently of growth factors and RAS as the sea urchin KSR3 does. Conversely, the spatial expression pattern of *ksr3* family members in various embryos may be used to predict previously unrecognized territories of ERK activation. It will therefore be interesting to correlate these expression patterns with the spatial pattern of activation of ERK signalling.

## KSR3 factors define a RAS-independent branch of the ERK signalling pathway

In this study, we have provided several lines of evidence supporting the idea that KSR3 proteins constitute a family of conserved and constitutively active activators of RAF regulating RAS-independent

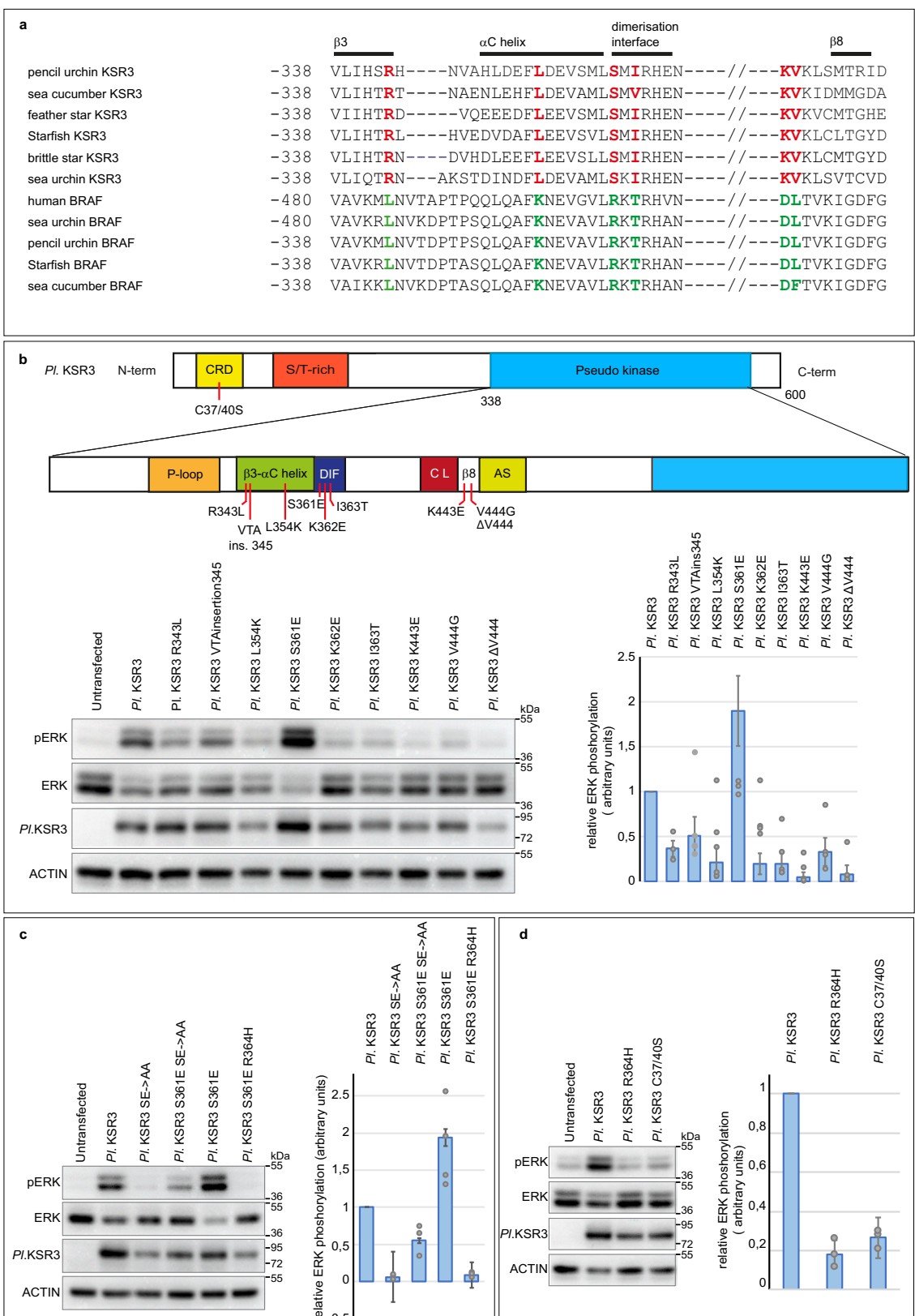

ERK signalling. First, we showed that KSR3 factors are clearly distinct from the other KSR proteins characterized so far. Unlike RAF and previously characterized KSR factors, KSR3 factors lack the N-terminal CC-SAM domain and possess a highly rearranged ATP-binding site and catalytic loop leaving little doubt regarding their lack of catalytic activity. Second, our biochemical experiments on ERK activation in human cells with the sea urchin, hemichordate and cnidarian KSR3 revealed that unlike the fly, nematode or vertebrate KSR factors, which do not activate ERK signalling when overexpressed and which have been implicated exclusively in RAS-dependent signaling[5,24–26], KSR3 factors possess an unusual intrinsic ability to activate MEK/ERK signalling in a RAS-independent but RAF-dependent manner due to the

**Fig. 7 | Substitution of Ser 361 by Glu in the dimerization interface of KSR3 creates a potent allosteric activator of RAF that partially bypasses the requirement for a phosphorylated NtA region. a** Alignment of KSR3 and RAF sequences from six different echinoderms. The sequence of human B-RAF is included in the alignment. Residues in red are present only in KSR3 but absent from RAF. These amino acid positions have evolved to become the donor specific residues of KSR3. Residues in green are present only in RAF but absent from KSR3. A set of conserved residues in the beta3 and beta8 strands, alpha C helix, and dimerization interface of KSR3 proteins as well as a shorter beta3/alpha helix distinguish KSR3 from B-RAF family members. **b** effects of individual substitutions of the donor specific residues in sea urchin KSR3 (Pl KSR3) on its ability to activate ERK signalling. The scheme describes the structure of KSR3 and the positions of the different mutations analysed. The graph represents the relative ERK activity of the mutants compared to Pl KSR3. Data represent the mean +/- s.e.m. of three independent transfections. **c** Substitution of Ser 361 of KSR3 by Glu creates an overactivated KSR3. The S361E mutation partially rescues the ability to activate ERK signalling to a mutant KSR3 lacking phosphorylation of the NtA motif. The graph represents the relative ERK activity of the mutants compared to Pl KSR3. Data represent the mean +/- s.e.m. of three independent transfections. **d** effect of mutation of the C1 domain on the ability of KSR3 to activate ERK signalling. Mutation of the two cysteines of the C1 domain of KSR3 into serine reduces its ability to activate ERK to the same extent as a Pl KSR3 R364H. The graph represents the relative ERK activity of the mutants compared to Pl KSR3. Data represent the mean +/- s.e.m. of three independent transfections. Source data are provided as a Source Data file. Information about quantification and replication of the western blots are provided in Supplementary Table 1.

presence of several activating mutations in key regions of the proteins such as the P-loop, the β3 and β8 regions, the alpha C helix and the dimerization interface. In contrast, single overexpression of vertebrate, fly or nematode KSR factors is not sufficient to activate ERK in cell culture[7] and even overexpression of a KSR1 protein carrying a mutation that stabilizes its active closed conformation is not sufficient to activate ERK signalling[8,29]. This comparison highlights the fundamental difference of activities between the family of KSR factors that we have uncovered and the family of KSR proteins that had been characterized previously.

Using phylogenetic analysis, we showed that *ksr3* genes constitute a unique family of orthologous *ksr* genes that is widely represented in the genomes of a broad spectrum of organisms belonging to different animal phyla including protostomes and deuterostomes and extending from cnidarians up to hemichordates. Our analysis revealed that although vertebrates and nematodes do have a pair of *ksr* genes named *ksr1* and *ksr2* in their genomes, these *ksr1* and *ksr2* genes are paralogous genes that evolved from a recent duplication of the *ksr1* gene. Therefore, the *ksr3* gene appears, which was present in the common ancestor of protostomes and deuterostomes has been lost secondarily in chordates as well as in insects and nematodes, which probably explains why no *ksr3* family member has been recovered in genetic screens looking for regulators of MAPK signalling conducted in these organisms.

**Why have chordates lost *ksr3* and RAS-independent signalling?**
Evolution of chordates has been accompanied by the amplification of several families of RTK ligands as shown in the FGF family which includes 33 members in the human genome. Perhaps the need for a cell-autonomous mechanism of activation of ERK signalling became less critical in these organisms compared to other organisms less well equipped with RTK ligands such as the sea urchin, which appears to possess only a handful of FGF ligands in its genome. Alternatively, genes like *ksr3* may have been counter-selected in chordates due to deleterious effects possibly caused by spurious activation of ERK signalling resulting from bursting and stochastic expression of *ksr3* in the course of development. Finally, it is intriguing to see that *ksr3* was lost from the genomes of vertebrates while the *raf* family of genes was expanding during the two rounds of genome amplification that took place soon after the emergence of vertebrates. Did the amplification and diversification of the *raf* gene rendered the presence of the *ksr3* gene superfluous? Although this hypothesis is interesting it is unlikely first because *ksr3* was also lost in organisms that had only one copy of the *raf* gene (like the fly or *C. elegans*). Another reason why the two events are probably not linked is that the loss of *ksr3* occurred in the common ancestor of chordates, i.e. before the large-scale genome amplification that led to the appearance of the three *raf* genes in vertebrates.

**RAS-independent activation of ERK as the ancestral mode of ERK signalling?**
Our findings suggest that expression of a constitutively active KSR3 factor constitutes an alternative RAS-independent and cell-

autonomous mode of activation of ERK signalling in non-chordate metazoa (Fig. 10). The widespread distribution of this mode of ERK activation signalling through a large portion of the animal kingdom (with the notable exception of insects, nematodes and chordates) is evocative of an evolutionarily ancient mechanism possibly used to activate ERK signalling independently of growth factors. The fact that *ksr3* orthologs are present in basal metazoans such as placozoa raises the intriguing possibility that this cell-autonomous mechanism of ERK activation may have predated the emergence of multicellular organisms and preceded the invention of the RTK-dependent and RAS-dependent signalling modules. In this scenario, the RAS independent way to activate ERK would have been the ancestral mechanism used to activate ERK and the requirement for ligands, RTK and RAS would have emerged secondarily. However, we did not detect the presence of *ksr3* orthologs in choanoflagellates, which are equipped with a bona fide RTK and we did not detect *ksr3* in the genome of sponges and ctenophores. The absence of *ksr3* in ctenophores and sponges suggests that *ksr3* arose after the emergence of these phyla. Nevertheless, the conservation of the cell-autonomous mechanism of ERK activation in a wide variety of extant species suggests an important role during evolution, possibly through functions of ERK during embryonic development and in the adult.

**Insights into the mechanism of activation of B-RAF**
Although our study focused on KSR3 from sea urchin, it provided insights into the mechanism of activation of the human B-RAF kinase. Using the sequences of constitutively active KSR3 proteins as templates we identified several mutations that significantly increase the catalytic activity of human B-RAF. These mutations, which had never been described before, affect various key regions of the protein including the beta 3 and beta 8 folds, the alpha C helix and the dimerization interface. For most of these mutations it could not be easily predicted how they would affect B-RAF activity. We note that the two mutations of B-RAF that most strongly activate ERK signalling that we found, L485R and R506E, are, like the V600E mutation, largely independent of the presence of the R509H dimerization blocking mutation on the same protein[66]. This could suggest that these mutants may bypass the requirement for dimerization and therefore the requirement for the presence of R509 in the dimerization interface by adopting a stable and active conformation as monomers. Alternatively, the relative independence of the mutants to the presence of the R509H mutation could suggest that the affinity of the dimers carrying these mutations is significantly higher than that of RAF WT and therefore that the mutant dimers are less affected than the WT dimer by the presence of the R509H mutation[66]. Our finding that the B-RAF R506E mutation is more prone to dimerization than WT B-RAF is consistent with this latter hypothesis.

One of the strongest mutations of B-RAF that we found is L485R. L485, on strand β3, is part of hydrophobic network that comprises F497 on the helix alpha C, L597 and V600 on the AS[9]. In EGFR, mutation of L834R (also called L858R) of the activation segment results in a

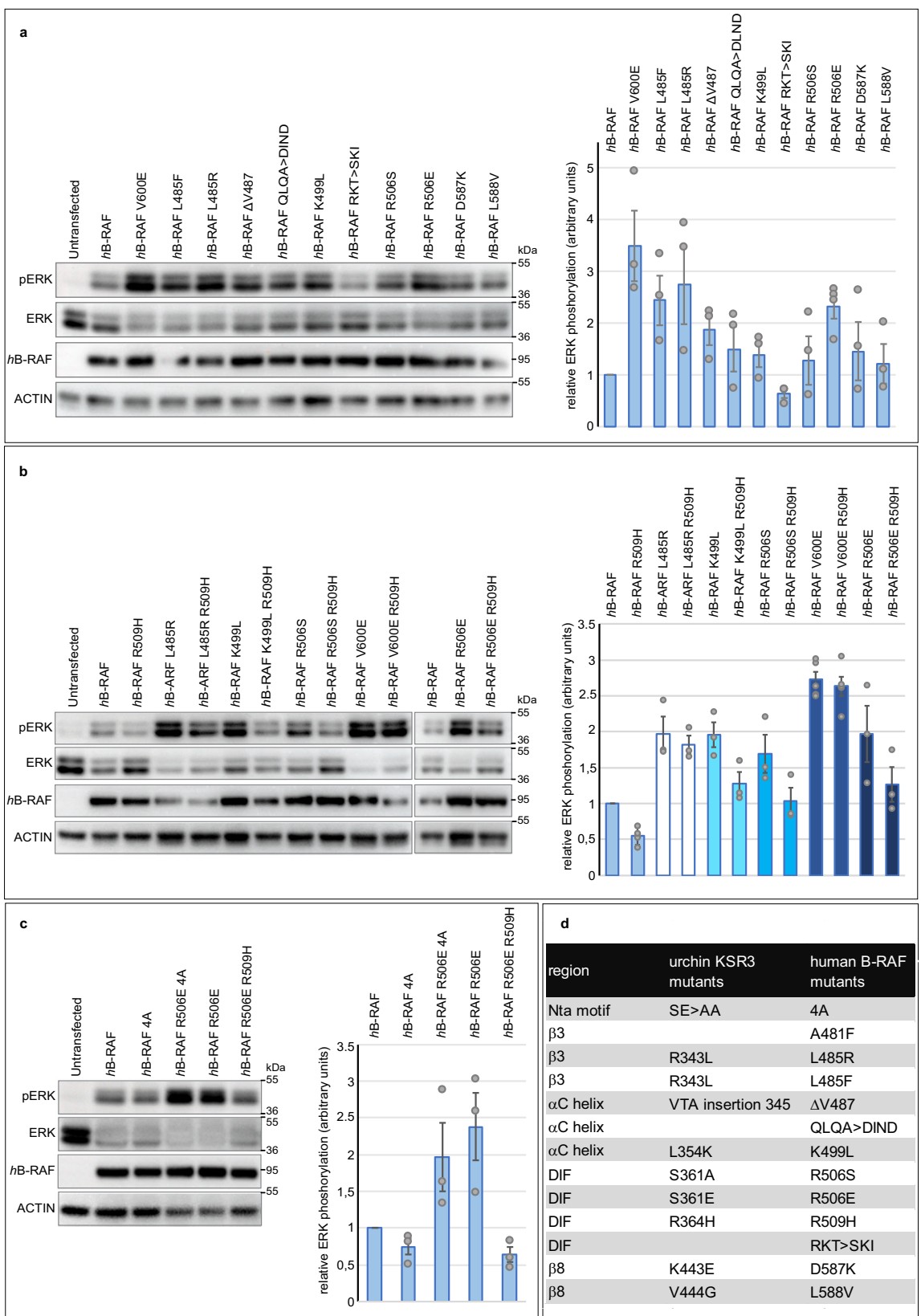

20-fold stimulation of its kinase activity and has been shown to cause oncogenic activation[67]. It is therefore likely that L485R of BRAF functions like L834R of EGFR, by disrupting the hydrophobic network that maintains the "off" conformation of the kinase. While the L834R mutation of EGFR is relatively frequent[67], for unknown reasons, the L485R mutation of B-RAF appears to be much less frequent.

The R506E mutation of B-RAF is also interesting first, because it has a strong impact on the activity of the protein and also because it is located in a strategical position within the dimerization interface at the basis of the alpha C helix. Finding that a mutation in the DIF of B-RAF results in overactivation of its catalytic activity was unexpected since previous work had shown that integrity of the DIF is

**Fig. 8 | The sequence of the constitutively active KSR3 can be used to predict mutations that over-activate human B-RAF. a** Effects of individual mutations of B-RAF on its ability to activate ERK signalling. The graph represents the relative ERK activation caused by the mutants compared to WT human B-RAF. Data represent the mean +/− s.e.m. of three independent transfections. **b** Like the V600E, the strongest novel gain-of-function mutants of B-RAF such as L485R and R506E are only partially affected by the presence of the R509H mutation. The two blots are from different experiments but contain the appropriate controls. **c** Like the S361E

mutation of KSR3, the R506E activating mutation of hB-RAF rescues the ability to activate ERK signalling to a mutant form of hB-RAF in which the SSDD motif of the NtA has been replaced by AAAA. The graph represents the relative ERK activity of the mutants compared to WT human B-RAF. Data represent the mean +/− s.e.m. of three independent transfections. **d** Correspondence between the residues mutated in the sea urchin KSR3 protein and the homologous residues in human B-RAF. Source data are provided as a Source Data file. Information about quantification and replication of the western blots are provided in Supplementary Table 1.

crucial for the activity of B-RAF[66]. The C-terminal end of the alpha C helix of B-RAF is markedly basic and contains three positive charges in the RKTR motif and it has been shown that substitution of these positively charge residues by alanine impairs activity[7,68]. Embedded within the dimerization interface, these positive charges likely exert a repulsive action on the RKTR motif of the other protomer[11]. In contrast, the C-terminal end of the alpha C helix of KSR3 factors is much more neutral, with the SMIR motif being the most prevalent within different phyla, suggesting that the presence of positively charged residues at the basis of the alpha C helix as in vertebrates, is actually not mandatory for the activity of B-RAF/KSR factors. Indeed, as predicted, substitution of R506 of B-RAF by a serine (a residue present on most KSR3s), increased significantly its activity confirming that not only the basic nature of the amino acid in position 506 is not crucial but it is not optimal for the activity of the protein. We found that, as in the case of KSR3, in which substitution of S361 by E increases its dimerization and transactivation potential, substitution of R506 of human B-RAF by a glutamic acid over activated B-RAF and increased its potential for dimerization and transactivation. Recent MD studies using RAF dimers phosphorylated on the NtA regions have implicated R506 in formation of intermolecular salt bridges with the phosphorylated NtA motif of the other protomer[11]. Also, previous studies had described oncogenic mutations of B-RAF consisting of insertions of three residues (VLR) at the level of R506, highlighting the key role played by this amino acid in the DIF[69,70]. Based on the structure of B-RAF homodimers published by Jambrina and coll., it is tempting to speculate that in the B-RAF R506E/B-RAF WT dimers, the negatively charged side chain of E506 of the activated B-RAF mutant interacts with the positively charged side chain of R506 on the other protomer creating a salt bridge with R506. This salt bridge, which would mimic the interaction between R506 of the receiver kinase and the phosphorylated NtA of the donor kinase, would act as a molecular clamp, linking the two DIF and connecting the alpha C helices of the two protomers, strongly stabilizing the dimerization interface by electrostatic interactions and/or hydrogen bonds. In other words, this salt bridge would increase the affinity of the two protomers and would promote transactivation. This mechanism would explain why, in the B-RAF dimer, the R506E mutation present on the DIF of one protomer allosterically promotes transactivation of the other protomer even in the absence of a phosphorylated NtA motif.

### Mechanism of the KSR3 S361E and B-RAF R506E activating mutations

At the molecular level, several mechanisms leading to hyperactivation of B-RAF have been described, including phosphorylation or introduction of negative charges in the activation segment as in the V600E mutant[44], shortening of the alpha C helix as in the Delta NVTAP mutant[44,53,69], in frame insertion in the alpha C beta loop as in the R506[insVLR][69,70], increasing the hydrophobicity of the alpha C helix, as in the L485F mutant[9] and increasing the hydrophobicity of the third R-spine residue, as in the L505F mutant[9]. The KSR3 S361E and B-RAF R506E mutations do not enter in any of these categories since unlike the vast majority of high activity mutations, they do not map to the P-loop or to the activation segment and do not correspond to insertions or deletions and therefore, they may represent an undescribed

mechanism of activation of B-RAF/KSR3. We propose that the KSR3 S361E and B-RAF R506E mutations that we have identified mimic the interaction between the phosphorylated NtA motifs on one protomer and R364 (KSR3) or R506 (B-RAF) on the other protomer, causing assembly of the active conformation of this protomer.

In conclusion, this study on the mechanism of activation of the catalytic activity of B-RAF by KSR3 factors emphasizes the central role of this newly described family of allosteric regulators of B-RAF in non-chordate metazoa.

In multicellular organisms, ERK relays information from RTKs and therefore it is widely believed that the key determinant for the time and place of activation of MAPK is the expression of active ligands. The RAS and RTK-independent mechanism of ERK activation by *ksr3* expression that we described invites to refine this view: at least in organisms that possess the *ksr3* gene, in addition to expression of RTK ligands, activation of ERK signalling can also be triggered by the regulated expression of natural, constitutively active regulators of RAF such as KSR3.

## Methods

### Animals, embryos and treatments

Adult sea urchins (*Paracentrotus lividus*) were collected in the bay of Villefranche-sur-Mer and Saint Jean Cap Ferrat. Embryos were cultured as described in Lepage and Gache (1989, 1990)[71,72]. Fertilization envelopes were removed by adding 1 mM 3-amino-1,2,4 triazole (ATA) 1 min before insemination to prevent hardening of this envelope followed by filtration through a 75 μm nylon net.

### Immunostaining

Embryos were fixed with 4% formaldehyde for 15–20 min then briefly permeabilized with methanol. Anti-Phospho-p44/42 MAPK (Erk1/2) (Thr202/Tyr 204) was used at 1/700. Embryos were imaged with an Axio Imager M2 microscope.

### In situ hybridization

Probes derived from pBluescript vectors were synthesized with T7 RNA polymerase after linearization of the plasmids by NotI, while probes derived from pSport were synthesized with SP6 polymerase after linearization with XmaI.

In situ hybridization was performed using standard methods with DIG-labelled RNA probes and developed with NBT/BCIP reagents.

Control and experimental embryos were developed for the same time in the same experiments. Embryos were imaged with an Axio Imager.M2 microscope. The *msp130* and *deadringer* probes have been described[12,73].

### QPCR

QPCR was performed as described previously[74] on a StepOne instrument. *cyclin-T* was used as a reference gene[75]. RNA was extracted using Trizol and treated with DNaseI before reverse transcription. cDNA synthesis was performed using a mixture of random and anchored oligo-dT20 primers. The relative expression levels of *P. lividus ksr3* and *pmar1* were calculated using a classical comparative Ct method Fold change = ($2^{-(dCtgene-dCtref)}$). The expression in the egg cell-stage was taken as 100%. Two biological replicates were used in this experiment.

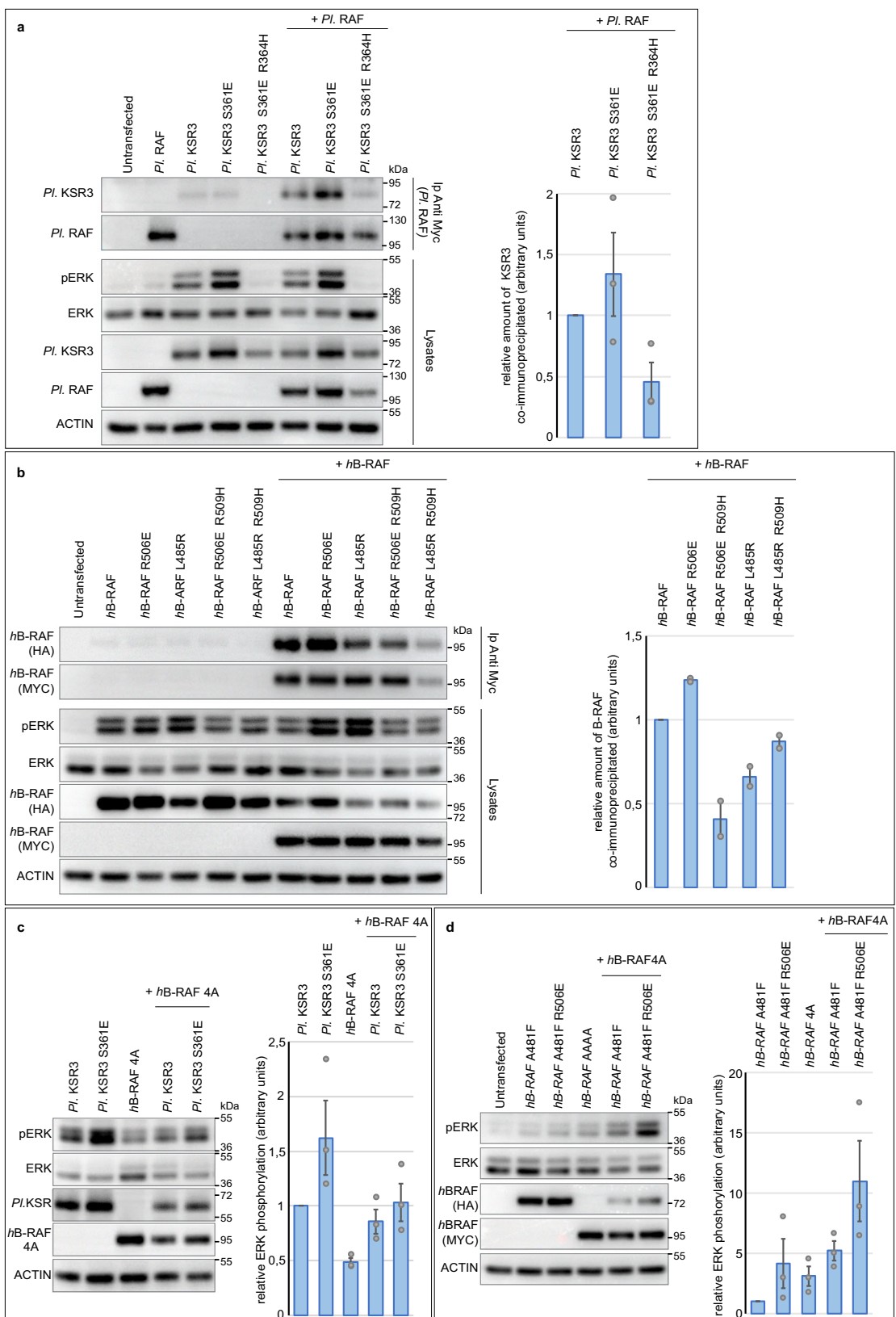

**Fig. 9 | The constitutively active KSR3 S361E and B-RAF R506E mutants activate ERK signalling both by promoting dimerization and by strengthening trans-activation of the B-RAF receivers. a** Co-immunoprecipitation of sea urchin RAF and wild-type or mutant KSR3. **b** Co-immunoprecipitation of human B-RAF with wild-type or mutant R506E or L485 activated B-RAF mutants. **c** Transactivation assay with the wild type KSR3 or the activated KSR3 S361E mutant. The presence of the S361E mutation on KSR3 increases its ability to activate ERK signalling through a B-RAF AAAA receiver. **d** Transactivation assay with a kinase-defective form of B-RAF or with the activated B-RAF R506E mutant. The kinase-defective B-RAF activator weakly transactivates a B-RAF receiver but a kinase-defective B-RAF activator carrying the activating R506E mutation strongly transactivates the B-RAF receiver. Data represent the mean +/− s.e.m. of three independent transfections. Source data are provided as a Source Data file.

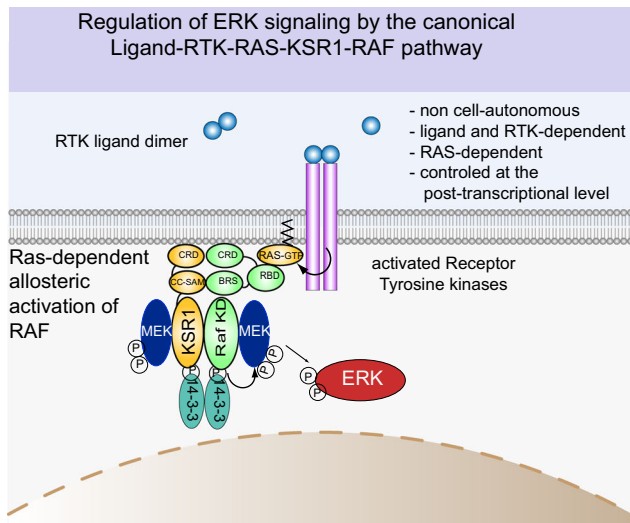

**Fig. 10 | An ancient mechanism for activation of ERK in non-chordate metazoa independent of cell interactions, RTK and RAS and model for KSR3 function.** Left panel, the canonical mechanism of RAS-dependent activation of ERK, relies on binding of ligands to RTKs, which in turn activate RAS. In this mechanism the function of phosphorylated KSR1 factors is to potentiate activation of RAF in a RAS-dependent manner. The CC-SAM domain of KSR1 and the B-RAF specific domain (BRS) of RAF interact and allow dimerization. Right panel, in contrast, the mechanism of KSR3-mediated activation of ERK signalling is independent of cell interactions and does not require ligands or RTKs. In this non-canonical mechanism, the function of KSR3 is to trigger strong activation of the pathway by directly transactivating RAF. Importantly, activation of ERK by KSR3 does not depend on RAS but critically requires dimerization with RAF. The activity of KSR3 appears to be regulated in a way similar to that of RAF and of other KSRs by interaction with 14-

3-3 proteins at the N-terminal (inhibitory) and C-terminal (activating) regions of the proteins. KSR3 is assumed to be in a dimerization competent conformation due to several sequence determinants such as the shorter alpha C helix, the presence of constitutively acidic NtA region and the presence of specific residues in key regions of the protein. After dephosphorylation of Ser87 of the Nta followed by phosphorylation of Ser586 and interaction of 14-3-3 proteins with the C-terminal region, KSR3 may transactivate RAF by inducing assembly of its active conformation. Note that in this model, activation of RAF occurs independently of RAS. The RAF-KSR3 dimer may anchor to the membrane only through the N-terminal cysteine-rich domains (CRD) of RAF and KSR3 but without interaction between RAS and the RAS binding domain (RBD) of RAF since RAS is in an inactive GDP bound form in absence of activation of RTK.

## QPCR Oligonucleotides used

*pmar1/hbox12*-fwd:5′-ATGATGGCGGATTCCACCATCATCACCCAAGTCTT

*pmar1/hbox12*-rev:5′-TTCTTGATGGGTTGACGAATAGA

*ksr3*-fwd:5′- CAGTTTGTCGGCACTTGCTA

*ksr3*-rev:5′- GATACGAGCTGCCGTGTGTA

*cyclinT*-fwd:5′-ACATGATGCCAACAGGTTCC

*cyclinT*-rev:5′-CAGATGCATCAATGGTGGATAA

## RNA isolation and RNA sequencing

In the RNA sequencing experiment, two independent batches of embryos were used for each condition. Embryos from control cultures (uninjected) or embryos injected with *pmar1* mRNA or with *dn-cadherin* mRNA were collected at the early blastula stage (6h30 after fertilization at 18 °C) and RNA was purified using Trizol. Libraries were made and sequenced by the GATC company. Briefly, polyadenylated transcripts were purified by two rounds of oligo-dT chromatography and cDNA libraries were constructed using a Superscript double stranded cDNA synthesis kit (Invitrogen), fragmented by sonication, and cDNA fragments of 200–500 bp were gel purified. After addition of Illumina adapters to these fragments, the libraries were sequenced on an Illumina HiSeq 2000 platform with 2 × 100 bp paired end sequencing. After sequencing, adapter sequences were removed and low-quality reads were eliminated. In parallel, a reference transcriptome (180,000 transcripts) was obtained using Oases by assembling reads generated by sequencing libraries from a mix of embryonic and larval stages (unfertilized egg, 16 cell-stage, early blastula, swimming blastula, mesenchyme blastula, early gastrula, late gastrula, prism stage, pluteus (48 h), 6-arm pluteus, 8-arm pluteus). The open reading frames of this reference transcriptome were extracted to generate a reference "orfome". The trimmed and clean RNA reads were mapped against the reference orfome using Bowtie and the table counts was generated

with Cufflinks. Transcripts counts were normalized and processed using the Cuffdiff package to determine the expression levels and the genes differentially expressed in the different samples. In a second step, the reads were re-mapped against a *Paracentrotus lividus* reference genome (version PQN3S) (*Paracentrotus lividus* genome sequencing consortium to be published) and a differential expression analysis was performed with EdgeR. The results of this second analysis largely confirmed the results obtained by mapping against a reference transcriptome and are those presented in Fig. 2b.

## Phylogenetic analysis

RAF and KSR sequences from deuterostomes (vertebrates, cephalochordates, hemichordates and echinoderms), protostomes (arthropods, priapulids, molluscs, annelids) and from diploblastic phyla (placozoa, cnidarians, sponges) were recovered from Genebank (http://www.ncbi.nlm.nih.gov/) using well characterized orthologs of vertebrate RAF and KSR sequences and with the sea urchin KSR3 sequence. The list of accession numbers of the 65 sequences is provided below. Full-length sequences were aligned using Clustal Omega with default parameters (http://www.ebi.ac.uk/Tools/msa/clustalo/). Trees were built using the Maximum Likelihood method based on the Whelan and Goldman model[76] available at http://phylogeny.lirmm.fr/phylo_cgi/index.cgi. The tree was calculated using IQ-TREE using the default parameters. A consensus tree with 50% cut off value was derived from 1000 replicates of ultrafast Bootstrap approximation[77,78]. Numbers above branches represent bootstrap values, calculated from this consensus.

## Accession numbers of RAF and KSR sequences used in the phylogenetic analysis

Most sequences were retrieved from Genebank or SwissProt using the *Paracentrotus lividus* RAF and KSR sequences as queries.

Sequences of KSR and RAF transcripts from *Eucidaris tribuloides* and from *Parastichopus parvimensis* were retrieved from Echinobase (http://bouzouki.bio.cs.cmu.edu/Echinobase/Blast/PpBlast/blast.php) using Blast and sea urchin sequences as query. Sequences from *Schmidtea mediterranea* were retrieved from the PlanMine web resource (http://planmine.mpi-cbg.de/planmine/report.do?id=3065582). The *Clytia hemispherica* sequences were retrieved from the MARIMBA data base (http://marimba-test.obs-vlfr.fr)

*Homo sapiens* KSR3: Q6VAB6
*Homo sapiens* KSR1: Q8IVT5
*Homo sapiens* A-RAF: P10398
*Homo sapiens* B-RAF: P15056
*Homo sapiens* C-RAF: P04049
*Branchiostoma floridae* KSR1: XP_002587595.1
*Branchiostoma lanceolatum* RAF (partial): B2BLV3
*Saccoglossus kowaleskii* B-RAF: XP_006821654
*Saccoglossus kowaleskii*: KSR1: XP_006821895.1
*Saccoglossus kowalevskii* KSR3: XM_006825955.1
*Ptychodera flava* KSR3: GenBank: GDGM01459026.1
*Danio rerio* KSR1: XP_009303858.1
*Danio rerio* KSR2 XP_009299513.1
*Danio rerio* BRAF: NP_991307.3
*Schmidtea mediterranea* KSR3: transcript SMEST026170001.1 *S. mediterranea* S2F2
*Schmidtea mediterranea* KSR1 *Transcript:* SMEST029462001.1 *S. mediterranea* S2F2
*Schmidtea mediterranea* RAF *Transcript: dd_Smed_v6_10148_0_1 S. mediterranea*
*Drosophila melanogaster* KSR1: Q24171
*Rhipicephalus sanguineus* KSR3: XP_037511753.1
*Rhipicephalus sanguineus* KSR1: XP_037513097.1
*Rhipicephalus sanguineus* RAF: XP_037518642.1
*Lottia gigantea* KSR3: XP_009044189.1
*Lottia gigantea* RAF: XP_009053549.1
*Crassostrea gigas* KSR1: XP_034336330.1- (https://www.ncbi.nlm.nih.gov/protein/1843147679)
*Crassostrea gigas* KSR3: XP_034325584.1- (https://www.ncbi.nlm.nih.gov/protein/1843080189)
*Crassostrea gigas* RAF: XP_034306751.1
*Lingula anatina* KSR3: XP_013397833.1
*Priapulus caudatus* KSR3: XP_014664712.1-
*Priapulus* KSR1 (partial): XM_014809390
*Priapulus caudatus* BRAF: XP_014673683.1
*Trichoplax adherens* BRAF: XP_002112616.1
*Trichoplax adherens* KSR3: XP_002110813.1-
*Trichoplax adherens* KSR1:XP_002115730.1- (https://www.ncbi.nlm.nih.gov/protein/196011734)
*Amphiura filiformis* KSR3 partial: GenBank: GGVU01109015.1
*Aneissia japonica* KSR3:XP_033117077.1
*Anneissia japonica* KSR1:XP_033119348.1
*Limulus polyphemus* KSR3: XP_022249006.1-
*Limulus polyphemus* KSR1: XP_022249988.1- (https://www.ncbi.nlm.nih.gov/protein/1238872070)
*Limulus caudatus* BRAF: XP_022245256.1
*Limulus polyphemus* KSR3-isoformB XP_013784198.2
*Strongylocentrotus purpuratus* RAF: XP_781094.4 (https://www.ncbi.nlm.nih.gov/protein/?term=RAF%20purpuratus)
*Strongylocentrotus purpuratus* KSR1: XP_030847715.1 (https://www.ncbi.nlm.nih.gov/protein/?term=KSR1+purpuratus)
*Strongylocentrotus purpuratus* KSR3 XP_780355.4
*Paracentrotus lividus* RAF: MW803455
*Paracentrotus lividus* KSR1: MW803454
*Paracentrotus lividus* KSR3: MW691984

*Parastichopus parvimensis* KSR1: transcrit_25265
*Parastichopus parvimensis* KSR3 (partial): transcript74/76
*Parastichopus-parvimensis* RAF: _8970_Transcript_11/15
*Patiria miniata* KSR1: PMI_005311.1
*Patiria miniata* KSR3: PMI_015081.1
*Patiria miniata* RAF (partial): PMI_004711.1
*Eucidaris tribuloides* KSR1: Locus_23325_Transcript_1/1
*Eucidaris tribuloides* KSR3: Locus_19352_Transcript_1/1
*Eucidaris tribuloides* RAF (partial): Locus_12655_Transcript_1/1
*Capitella teleta* BRAF: accession KB308571.1
*Capitela teleta* KSR3: ELT92102.1
*Capitela teleta* KSR1: ELU07085.1
*Caenorhabditis elegans* KSR1:NP_509396.1
*Caenorhabditis elegans* KSR2:NP_001021518.1
*Caenorhabditis elegans* RAF:AAA28142.1
*Acropora millepora* KSR1-like: XP_029179528.1
*Acropora millepora* KSR3: XP_029213160.1
*Acropora millepora* A-Raf-like:XP_029212189.1
*Nematostella vectensis* RAF: EDO47089.1
*Nematostella vectensis* KSR3: XP_032227537.1
*Nematostella vectensis* KSR1: XP_032220394.1
*Clytia hemispherica* RAF:XLOC000587
*Clytia hemispherica* KSR1:XLOC036045
*Clytia hemispherica* KSR3:XLOC:029804
*Amphimedon queenslandica* RAF: XP_019850658.1
*Amphimedon queenslandica* KSR1:XP_019848730.1-
*Platynéreis dumerilii* KSR3: GenBank: HALR01292027.1
*Ramazzottius varieornatus* KSR3: GAU99520.1
*Ramazzottius varieornatus* RAF: GAU93787.1
*Hypsibius dujardini* KSR1: OQV18786.1

**Cloning of *ksr1* and *ksr3* sequences and construction of expression vectors**

cDNAs encoding KSR1, KSR3, RAF and RAS were retrieved from an EST cDNA library. The full-length cDNAs were sequenced entirely and the open reading frame was amplified using the oligonucleotides and the PCR product was digested and cloned into the pCS2 vector.

Oligonucleotides for making the pCS2 *ksr1* and *ksr3* construct are:
KSR3 BglII ATG FW: 5′- CCCAGATCTACCATGCAACCAAAGGAACATATAGC-3′
KSR3 EcoRI TAG REV:5′- CCCGAATTCTCACCGAAAGAGTCCGGTTCCGAG-3′
KSR3 EcoRI ATG FW: 5′- AGGGAATTCACCATGTCTGGAGATTGTTTATCGTC-3′
KSR1 XhoI REV: 5′- CCTCTCGAGTCATAACACAGGTTCAGCTGATC-3′
RAF-EcoRI FW 5′- GGCGAATTCACCATGGGAGGTCCGACGTCTGGAC-3′
RAF-XhoI REV 5′- TAGCTCGAGTCATTGACGCATGTCAAAAGG-3′

The dn-RAS construct that we used is derived from a human Ha-Ras carrying the Ser17Asn[79]. The open reading frame of the original clone in pSP64T was first amplified with oligos carrying restriction sites and cloned into pCS2 using the following oligos:
Ha-RAS BamHI ATG:
5′- ACC GGA TCC ACC ATG TAC CCA TAC GAT GTT CCA GAT TAC GCT GGC ATG ACG GAA TAT AAG CTG GTG G-3′
Ha-RAS EcoRI TGA: 5′-ACC GAA TTC TCA GGA GAG CAC ACA CTT GCA GCT-3′

Then two HA tags were inserted on the N-terminus using the following oligo:
dn-RAS 2HA fw:5′- ACC GGA TCC ACC ATG TAC CCA TAC GAT GTT CCA GAT TAC GCT GGC TAC CCA TAC GAT GTT CCA GAT TAC GCT GGC ATG ACA GAA TAC AAG CTT GTG GTG GTG GGC GCT-3′ and a reverse primer in the pCS2 sequence:
pCS2 reverse: 5′-GCT CCG ACG TCC CCA GGC AGA AT-3′

2HA DN-RAS (N17) FW: 5′- ACC GGA TCC ACC ATG TAC CCA TAC GAT GTT CCA GAT TAC GCT GGC TAC CCA TAC GAT GTT CCA GAT TAC GCT GGC ATG ACA GAA TAC AAG CTT GTG GTG GTG GGC GCT -3′

The dn-RAF construct is derived from the C-RAF S621A[80] by the addition of 2 MYC tags using the oligos:

C-RAF BamHI FW: 5′- ACC GGA TCC ACC ATG GAG CAA AAG CTC ATT TCT GAA GAG GAC TTG AAT GAA ATG GAG CAA AAG CTC ATT TCT GAA GAG GAC TTG AAT GAA ATG GAG CAC ATA CAG GGA GC-3′

C-RAF TAG EcoRI REV: 5′- GGT GAA TTC CTA GAA GAC AGG CAG CCT CGG-3′

For making the *Paracentrotus* pCS2 RAF 6 MYC tags, the open reading frame of RAF was amplified and the digested PCR product was cloned into pCS2 6xMYC tags at the EcoRI site using the following FW oligo:

RAF- EcoRI FW: 5′-GGCGAATTCAATGGCGACTGGCAGTGAATTT-3′
RAF-XhoI Rev: 5′- TAGCTCGAGTCAAACTGCTGTGGAACCAC-3′

To make the 3HA pCS2 *ksr3* and 3HA pCS2 *ksr1*, three copies of the HA Tag were introduced by PCR using the pCS2 *ksr3* and pCS2 *ksr1* plasmids as templates and using the following oligos:

KSR3 3HA FW:-5′ TACCCATACGATGTTCCAGATTACGCTGGCTA TCCCTATGACGTCCCGGACTATGCAGGATATCCATATGACGTTC CAGATTACGCTATGCAACCAAAGGAACATATA-3′

The resulting protein sequence is (HA tags are underlined and bold sequence from KSR3): <u>MYPYDVPDYA</u><u>GYPYDVPDYA</u><u>GYPYDVPDYA</u> **MQPKEHI**

3HA KSR1 FW:5′- TACCCATACGATGTTCCAGATTACGCTGGCTA TCCCTATGACGTCCCGGACTATGCAGGATATCCATATGACGTTC CAGATTACGCTATGTCTGGAGATTGTTTATCGTC

The resulting protein sequence is (HA tags are underlined and bold sequence from KSR1): <u>MYPYDVPDYA</u><u>GYPYDVPDYA</u><u>GYPYDVPDYA</u> **MSGDCLSS**

The *ksr3* sequence from *Nematostella vectensis* was amplified from a mixture of cDNAs from various stages and cloned into pCS2 using the following oligos/

KSR3 NV ATG BamHI:5′- ACCGGATCCACCATGCAAGAGCCGTCT CACAGTTTCG-3′

KSR3 NV EcoRI rev:5′- ACCGAATTCCTATAGCGGCGCGGTGCTGA ATGACAG-3′

A HA tagged version was then amplified using the following oligo:

KSR3 NV3HA: ACCATGTACCCATACGATGTTCCAGATTACGCT GGCTATCCCTATGACGTCCCGGACTATGCAGGATATCCATATGA CGTTCCAGATTACGCTATGCAAGAGCCGTCTCACAGTTTC

All PCR reactions were made using the Phusion high fidelity DNA polymerases and the constructs were entirely verified by sequencing.

A 3HA tagged *Saccoglossus kowaleskii ksr3* cDNA was synthesized by Genescript using the sequence of the *ksr3* transcript retrieved from Genebank and cloned into pCS2 at the BamH1 and XhoI sites.

The dimerization mutants of KSR3 from *Paracentrotus, Saccoglossus* and *Nematostella* were synthesized using the following oligos (mutated codon arginine in bold):

KSR3 PL R364H fw: 5′- **CAT**CATGAGAACATCGCTCTCTTTATG-3′
KSR3 PL R364H rev: 5′- AATCTTGCTCAGCATGGCGACCT-3′
KSR3 NV R390H fw:5′- ATGATC**CAT**CACGAGAACGTCGTGTTG
KSR3 NV R390H rev: 5′- GCTCAGCAGGGACACTTCTTTG-3′
KSR3 SK R350H Fw: 5′- AGTCGTATC**CAT**CATGAAAATATTG AATTG
KSR3 SK R350H rev: 5′-GAGTTTCTTGAAGAAGTTGCTATACTC-3′

The dimerization mutant of KSR1 was generated with the following oligos:

KSR1 R608H fw: 5′-**CAT**CATGATTTTGTGGTTCTATTTATG3′
KSR1 R608H Rev:5′- TGTCATCCTGAAGAATGAAACCT

The activation loop mutants were constructed by replacing D452 and D455 into Alanine and by substituting Alanine for T449,T453, S454 and T457 using the following oligos:

KSR3 D452A D455A fw: ACTAGT**GCC**GTCACATGTCAA-3′
KSR3 D452A D455A rev: **AGC**AACACACGTGACAGATAG-3′
KSR3 AS TTST > AAAA fw:GACGTC**GCA**TGTCAAAGGACAAAC-3′
KSR3 AS TTST > AAAA rev:5′-**AGCAGC**ATCAACACA**CGC**GACAG ATAGCTTGACCTT-3′

The NtA region mutants were made by replacing S306 and E307 by alanine in the sea urchin KSR3 protein or D293 and E294 by alanine in the Saccoglossus KSR3 protein using the following oligos:

KSR3 PL S306E307 > A fw:5′- GGTCTC**GCTGCA**TGGACCATTCCAC ACGAA-3′
KSR3 PL S306E307 > A rev 5′- TGCTTTCCTAGCTGCGTAGTG-3′
KSR3SK D293,E294 > AA fw: 5′-AAATTA**GCTGCA**TGGAAAATCAAC TATGAT-3′
KSR3 SK D293,E294 > AA rev:5′-GCTATGAATGTCTTCACTTTCAT ACTG-3′

The C-terminal 14-3-3 binding site mutants of KSR3 from *Paracentrotus* and *Saccoglossus* were synthesized using the following oligos (mutated codon serine residue in bold)

KSR3PL S586A fw: 5′-**GCA**GAGCCGGATAACCTCAATC-3′
KSR3 PL S586A rev: 5′-CAGACTGTGCTTCATGTTTCC-3′
KSR3 SK S572A fw: 5′- **GCA**GAGCCGGATCAGTTGAAC-3′
KSR3 SK S572A rev: 5′- TTGACTGTGTTTTTTATGTAACGGC
KSR3 NV S618A fw: 5′- **GCG**GAACCGGAGAAGATTCAC-3′
KSR3 NV S618A rev: 5′- GTGACTTCCGGTGATTCGCTT-3′

## Primers for construction of KSR3 mutants

KSR3 VTA insertion345 fw: 5′-AAGTCGACCGACATCAATGATTTC
KSR3 VTA insertion 345 rev: 5′- **CGCGGTAAC**CGCGTTTCTAGT CTGGATGAGGAC
KSR3 L354K fw: 5′-GAGGTCGCCATGCTGAGCAAGATTCGA
KSR3 L354K rev: 5′-GTC**TTT**GAAATCATTGATGTCGGT
KSR3 I363T fw: 5′-CATGAGAACATCGCTCTCTTTATGGGC
KSR3 I363T rev: 5′-TCG**AGT**CTTGCTCAGCATGGCGACCTG
KSR3 S361E/K362E FW: 5′-CGACATGAGAACATCGCCTC
KSR3 S361E rev: 5′-AATCTC**GCT**CAGCATGGCGACCTC
KSR3 K362E rev: 5′-AAT**CTT**CTCCAGCATGGCGACCTCGTC
KSR3 R343L fw: 5′-AAGTCGACCGACATCAATGATTTCC
KSR3 R343L rev: 5′-CGCGTTT**AGA**GTCTGGATGAGGACTTCGCC
KSR3 V444G fw: 5′-**GGC**AAGCTATCTGTCACGTGTGTT
KSR3 deltaV444fw: 5′-AAGCTATCTGTCACGTGTGTTG
KSR3 deltaV444 rev: 5′-CTTTGACTCGAGGTAGACACATCT
KSR3 K443E fw: 5′-GTCAAGCTATCTGTCACGTGTGT
KSR3 K443E rev: 5′-CTCTGA**CTC**GAGGTAGACACATCT
KSR3 C37/40 S FW: **AGT**GATTTC**AGC**AAGAAGTCTATG
KSR3 C37/40 S REV: GCGTTTAGCGGGAGATACGAG

## Primers for constructions of B-RAF mutants

hB-RAF L485F fw: 5′-AATGTGACAGCACCTACACC
hB-RAF L485F rev: 5′-**AAA**CATTTTCACTGCCCACATCACC
hB-RAF L485R rev: 5′-**CCG**CATTTTCACTGCCCACATCACC
hB-RAF ΔV487 fw: 5′-ACAGCACCTACACCTCAGCAG
hB-RAF ΔV487 rev: 5′-ATTCAACATTTTCACTGCCAC
hB-RAF QLQA- > DLND fw: 5′-TTCAAAAATGAAGTAGGAGTACTC
hB-RAF QLQA- > DLND rev: 5′-**GTCATTTAAATC**CTGAGGTGTAG GTGCTGTCAC
hB-RAF K499L fw: 5′-TTACAAGCCTTC**TTG**AATGAAGTAGGAGTAC
hB-RAF K499L rev: 5′-CTGCTGAGGTGTAGGTGCTGTCAC
hB-RAF RKT- > SKI fw: 5′-CGACATGTGAATATCCTACTC
hB-RAF RKT- > SKI rev: 5′-**TATTTTACT**GAGTACTCCTACTTCATT TTTG
hB-RAF R506S fw: 5′-AAAACACGACATGTGAATATC

hB-RAF R506S rev: 5′-**GCT**GAGTACTCCTACTTCATTTTTGAAGG
hB-RAF R506E rev: 5′-**CTC**GAGTACTCCTACTTCATTTTTGAAGG
hB-RAF D587K fw: 5′-ACAGTAAAAATAGGTGATTTTGG
hB-RAF D587K rev: 5′-GAG**CTT**TTCATGAAGAAATATATTATTACTC
hB-RAF L588V rev: 5′-**GAC**GTCTTCATGAAGAAATATATTATTAC
hB-RAF R509H fw: 5′-GTGAATATCCTACTCTTCATGG
hB-RAF R509H rev: 5′-ATG**GTG**TGTTTTCCTGAGTAC
hB-RAF A481F rev: 5′-CAACATTTT**GAA**CACATC
hB-RAF NVTAP fw: 5′-ACACCTCAGCAGTTACAAGCC
hB-RAF NVTAP rev: 5′-CAACATTTTCACTGCCACATC
hRAS G12V rev: 5′-TTTACCAACACCTACAGC**TCC**

## The PP2A regulatory subunit mutant was made using the following oligos

pp2a reg sub XhoI fw: 5′-TAGCTCGAGACCATGGCCGTTTCAGGCGAGG TG3′ and pp2a reg sub XbaI rev: 5′- TTCTCTAGATTAGCTGGGTGGTT TCTCTTG-3′

**The PP2A catalytic subunit cDNA** was cloned using the following oligos:

PP2AC EcoRI fw: 5′-AGGGAATTCACCATGGAGGATACCAAAAA GGA-3′

PP2AC XhoI rev: 5′- CCTCTCGAGCTACAAGAAGTAGTCAGG CG-3′ and

The dominant negative PP2AC mutant was obtained using the following oligos:

PP2AC L209P fw: 5′-CCCTGGTCTGATCCAGACGACAGG-3′
PP2AC L209P rev: 5′- CAGGTCACACATAGGTCCTTC-3′

## Overexpression of mRNAs and morpholino injections

mRNA and morpholino injections were performed as described in Molina et al. (2019) (39) For overexpression studies, capped mRNAs were synthesized from NotI-linearized templates using mMessage mMachine kit (Ambion). After synthesis, capped RNAs were purified on Sephadex G50 columns and quantitated by spectrophotometry. RNAs were mixed with Tetramethylrhodamine Dextran (10,000 MW), Texas Red Dextran (70,000 MW) or Fluoresceinated Dextran (70,000 MW) at 5 mg/ml and injected in the concentration range 100–1000 μg/ml.

Wild type *pmar1* mRNA was injected at 10 μg/ml and *dn-cadherin* RNA at 500–750 μg/ml. Wild-type *ksr3* mRNA was injected at 1000 μg/ml and mutated *ksr3* mRNAs were injected at 500–1000 μg/ml.

The human *dn-ras* mRNA (RAS Ser17Asn) was injected at 600 μg/ml.

Morpholino oligonucleotides were dissolved in sterile water and injected at the one-cell stage together with Tetramethylrhodamine Dextran (10,000 MW) or Fluoresceinated Dextran (FLDX) (70,000 MW) at 5 mg/ml. For each morpholino a dose-response curve was obtained and a concentration at which the oligomer did not elicit non-specific defect was chosen. Approximately 2–4 pl of oligonucleotide solution were injected in the experiments described here. All the injections were repeated multiple times and for each experiment >25 embryos were analysed (see Supplementary Table 1 for more detail of the number of experiments and number of embryos analysed). Only representative phenotypes present in at least 80% of the injected embryos are presented.

## Antisense Morpholino oligonucleotides

Antisense Morpholino oligonucleotides were obtained from Gene-tools. Morpholino oligonucleotides were resuspended in sterile water and injected into fertilized eggs at different concentrations to determine the concentration range of oligo that did not cause toxicity. The sequences of the antisense morpholino oligonucleotides used in this study are the following

(ATG and intron sequence in bold):
KSR3-ATG: 5′-TGCTATATGTTCCTTTGGTTG**CAT**G-3′
KSR3-Splice: 5′-**GACGTTTTGTTGGAACTTAC**TTGAA-3′

Alx1-ATG: 5′- GCATGGAGTTAGGGTAGAACAA**CAT**-3′
Ets1-ATG: 5′- GGAAGGAGCAGTGCATAGATGC**CAT**-3′
HesC-ATG: 5′- GTAACCAGTTGAGGTAAG**CAT**GT-3′

The splice morpholino targets the exon4-intron4 sequence of *ksr3*.

## RNA extraction and cDNA synthesis for Mo ksr3 splice injected embryos

The Exon4 intron 4 splice site was selected to block ksr3 pre-mRNA splicing. The efficiency of this splice-blocking morpholino was monitored by semi quantitative RT-PCR. 200 embryos from control cultures (uninjected) or embryos injected with *ksr3* morpholino splice (0.6 mM) were collected at the swimming blastula stage (16 h after fertilization at 14 °C). RNA was extracted using Trizol and cDNA synthesis was performed using a mixture of random and anchored oligo-dT20 primers from New England Biolabs (S1330S).

After synthesis, 2 μL of each sample was used to perform a PCR with Gotaq® G2 Taq polymerase from Promega and primers located in exon 3 and exon 5.

Ksr3 exon 3 fw: 5′-CAATTTCTGTCCTCACATCCCTG-3′
Ksr3 exon 5 rev: 5′-CTGTTAACGACGGCGAGTTTG-3′

After gel electrophosresis, PCR products were purified, cloned in pGEM®-T Easy Vector Systems from Promega and sequenced.

## Accession numbers of *Paracentrotus lividus* mRNA sequences

Alx1: ABG00197
ETS1: AAS00537
HESC: GCZS01082823.1
RAF: MW803455
KSR1: MW803454
KSR3: MW691984
Pmar1/Hbox12: X83675.1

## Expression constructs and HEK293T cell transfections

For expression in HEK293T cells, the open reading frames of the desired constructs were cloned in the pCS2+ vector downstream of the CMV promoter. HEK293T cells were maintained in DMEM supplemented with 10% foetal bovine serum, penicillin and streptomycin at respectively 100 u/ml and 100 μg/ml at 37 °C and under 5% $CO_2$. About 500,000 cells were seeded on a 100 mm tissue culture plate 18 h before transfection. Cells were transfected using the cationic polymer Polyethylenimine (PEI) as described by Boussif et al.[81]. PEI (800 kDa) (Fluka) was dissolved in water at 0.9 mg/ml stock in H2O, adjusted to pH: 7 and filter sterilized. Immediately before transfection the plasmids (8 μg) were diluted into 1 ml of DMEM (without FBS, Penicillin or Streptomycin) then a three-fold amount of Polyethylenimine (24 μg) was added. After a 15 min incubation at room temperature, the transfection mixture was added to the plate of cells. A control plasmid coding for the Greeen Fluorescent Protein was used as control to estimate the efficiency of transfection. For co-transfections, 8 μg of each plasmid and a total of 48 μg of PEI was used for transfection and the medium was replaced with fresh DMEM 6 h after transfection. After 24 h, the cells were washed twice with Phosphate Buffer Saline (PBS) and lysed with 1 ml of lysis buffer (50 mM Tris HCl pH: 7.5, 150 mM NaCl, 10% Glycerol, 0.2% Triton X100, 1 mM EDTA, supplemented with protease and phosphatase inhibitors cocktails from Roche (ref 04693124001 and 04906837001) for 20 min at 4 °C. Lysates were centrifuged at 14000 g for 10 min at 4 °C and their protein concentration was measured using the Bradford assay (Biorad). The supernatants were kept at −80 °C.

## Immunoprecipitations

For immunoprecipitations, 200 μl of lysates at 0.5 mg/ml were mixed with 3 μl of mouse anti-myc antibody (9E10) and incubated at 4 °C

overnight with agitation. The next day, 25 μl of Dynabeads Pan mouse IgG were added and incubation was continued for 2 h at 4 °C. At the end of the incubation beads were washed five times with lysis buffer (50 mM Tris HCl pH: 7.5, 150 mM NaCl, 10% Glycerol, 0.2% Triton X100, 1 mM EDTA, supplemented with protease and phosphatase inhibitors cocktails) and bound proteins were eluted with 25 μl of sample buffer and heated 2 min at 95 °C. After electrophoresis and immunoblotting, anti-HA and anti-Myc antibodies from rabbit were used to detect the corresponding antigens.

## Western blotting

Protein samples (10–50 μg/lane) were separated by SDS-gel electrophoresis and transferred to PVDF membranes (0.5 μm). After blocking in 5% dry milk, blots were incubated overnight with the primary antibody diluted in 5% BSA in TBST. After washing and incubation with the secondary antibody, bound antibodies were revealed by ECL immunodetection using the SuperSignal West Pico Chemiluminescent substrate (Pierce) and imaged with a Fusion Fx7.

All the scans of uncropped blots are included in a source file.

## Antibodies used

-anti-Phospho-p44/42 MAPK(Erk1/2)(Thr202/Tyr204)(Cell signalling, D13.14.4E, ref.4370)

-anti p44/42 (ERK1/2) Rabbit Cell signalling 137F5)
-anti-HA high affinity (from Rat) (Roche, ref. 3F10)
-anti-HA (from Rabbit) (Cell signalling, ref 3724 T)
-anti-βActin (Cell signalling, ref. 4967)
-anti-Myc clone 9E10 from mouse (Thermofisher ref 13–2500)
-anti Myc (from Rabbit) (Cell signalling ref 2278 S).

## Reporting summary

Further information on research design is available in the Nature Portfolio Reporting Summary linked to this article.

# Data availability

All data described in the manuscript or the supplementary materials is available. The RNA-seq data has been deposited to ArrayExpress under the accession code E-MTAB-12938. Source data are provided with this paper.

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

## Acknowledgements

We thank Thomas Lamonerie for help with the cell cultures and Thomas Lamonerie, Nathalie Billon, Laurent Gagnoux and Christian Gache for careful reading of the manuscript. We thank Gilles Pagès, Philippe Lenormand, Roser Busca and Patrick Brest for providing us with reagents and for interesting discussions. We thank Julie Hanotel for help in the early phase of the project and for stimulating discussions and Sebastien Huault and Antoine Fortune for advice and interesting discussions. We thank Eric Röttinger for his generous gift of Nematostella cDNAs and François Lapraz and Ferdinand Marletaz for their precious help with the phylogeny. We thank Sarah Assaf for help with the contruction and testing of the CRD mutant, Evelyn Houliston for help to retrieve the Clytia KSR cDNA sequences and for a generous gift of cDNA. We thank Man-uela Baccarini (University of Vienna) for providing us with the human wt Braf as well as the B-RAF V600E and BRAF 4 A constructs.

This research was funded, in whole or in part, by the Foundation for Medical Research, (Grant DEQ20180339195) and by the Foundation for Cancer research (ARC) (Grant: ARCPJA32020060002217). A CC-BY public copyright license has been applied by the authors to the present document and will be applied to all subsequent versions up to the Author Accepted Manuscript arising from this submission, in accor-dance with the grant's open access conditions.

## Author contributions

Design of the experiments: TL, NDC, AC, MDM. Methodology: NDC, AC, TL, PD, LM, LTP. Investigation: NDC, AC, TL, MDM, LTP. Funding acqui-sition: TL. Project administration: TL. Supervision: TL. Writing – original draft: TL.

## Competing interests

The authors declare no competing interests.
