## [Peer Review File · Nature Communications]

RAS-independent ERK activation by constitutively active KSR3 in non-chordate metazoaREVIEWER COMMENTS

Reviewer #1 (Remarks to the Author):

This manuscript by Chessel et al. convincingly demonstrates that a constitutively active version of a gene related to vertebrate KSR1 and KSR2, termed KSR3, is necessary for the cell-autonomous, RTK-RAS independent, mechanism that activates p-ERK signaling in a subset of mesoderm cells downstream of the PMAR/HesC double negative gate. They go on to show that KSR3 is an evolutionarily conserved in most metazoans, excluding chordates. Then the authors go into detail to determine the role of the various domains of KSR3 and their interactions and activation of the RAF map kinase upstream of p-ERK. Finally, they use their data from the KSR3 mutations to identify novel putative mutations in human Raf that may have implications in human disease. Overall, this is a well written manuscript, and the experiments are thorough and well performed, and presented. This work will be of considerable interest to developmental biologist as well as scientists researching RTK/RAS/Raf signaling and its role in human disease and I recommend it for publication. However, there are a few issues that would improve the manuscript and should be addressed.

Major issues:

- For figure 1e, control embryos should be shown for each experiment.
- In the second section (Pg. 8, starting on line 196), the RNA-seq screen is not explained very well and could be confusing to readers. The authors should consider reworking the intro of this sections. Also, it could help if the authors created a better diagram of the RNA-seq screen in Fig. 2a instead of just showing the phenotypes of the perturbations?
- The KSR MO experiments require that specificity controls be performed. While they do use two morpholinos, there is no antibody available so they should do a standard rescue experiment with KSR mRNA co-injection. They use a splice blocking morpholino but don't show that its blocking correct RNA processing using a simple PCR assay. They should perform this experiment.
- The conclusions stated for the paragraph starting on pg. 13, line 309 are overstated. The data indicate that KSR3 is necessary for RAS-independent activation phospo-ERK, not "absolutely" required and sufficient. To show sufficiency the authors should inject on of the constitutively active construct into a non-mesodermal blastomere and assess Erk activation and phenotypes. Thus, the authors should consider either softening the language or performing the experiments.
- The functional experiments showing that mutating RAF dimerization sites in KSR1 and KSR3 are interesting, but the authors overstate their conclusions from these experiments. At this point in the story these experiments do not prove that KSR3 dimerizes with RAF and it is not clear how these results suggest that "KSR3 prevents allosteric transactivation of Raf by KSR3"? The authors should consider placing these experiments in Fig. 9 after they biochemical experiments showing dimerization between KSR3 and RAF. These functional in vivo results would strengthen the argument there instead of seeming out of place where they currently stand. Also, these experiments do not show the WT KSR3 overexpression phenotypes when it seems they should be included with these experiments and then a sentence or two about why they don't elicit a phenotype. Also, do the authors address why overexpressing KSR3 in sea urchins doesn't increase p-Erk whereas it does increase p-Erk in human cell lines (Fig. 6a)?
- The entire sections entitled "The ability of KSR3 proteins to activate ERK signaling..." on page 21 line 533 as well as the section entitled "KSR3 S361E..." on page 30 are hard to follow as written. A particular problem is that it is difficult to match the various constructs that they talk about to the experiments shown in Figure 6. They should better define the various constructs in the manuscript. Also, the authors place a KSR3 dimerization mutant experiment in figure 6e. Why not place this in

Figure 9. Again, this seems out of place like the previous dimerization experiment in Fig 3.

- The same critique could be applied, although to a lesser extent, for the other western blot/biochemical interactions sections. All these experiments are very well-done experiments, but the data are extremely dense, and it would be better on readers if the authors provided more context and made it clear which constructs that are talking about in the main body of the manuscript. For instance, the authors could indicate the abbreviation used in the figures in parentheses after they describe the mutant in the text. A diagram of the various mutations they use in each of these sections could be added to these figures for easier reference instead of going back and forth to figure 2 and figures 5-9. It's hard work.
- The discussion is interesting, but very long. The authors should consider reworking this section so that it is more succinct.

Minor issues:

- There are many grammatical errors throughout the manuscript, including several run-on sentences (e.g., the sentence on page 35, lines 817-832). Also, it is hard to determine where paragraphs start/end because there are no indentations at the beginning of paragraphs or spaces between them. The authors should perform a careful edit of the writing before resubmission.
- beta-catenin should be capitalized if talking about the protein or use the Greek Beta.
- Image in Fig1C for VEB ets1 is not clear. Is ets1 expressed ubiquitously at this stage?
- The writing in the section entitled "KSR is required for activation of ERK..." on page 13 is difficult to follow. In the first and second paragraphs of the section the authors should consider providing more context for the experiments. As is, these paragraphs read as a list of what they did.
- Were the HesC, Ets1, Alx1 and KSR1 morpholinos previously characterized? If so, then these morpholinos should be referenced.
- Line 312 p13 the sentence states "We confirmed these results with a second morpholino..." needs a figure indication showing the phenotype of these morphants.

Reviewer #2 (Remarks to the Author):

In this study, the authors first identified KSR3 as a transcriptional target of Pmar1 (via a double negative gating) in micromeres of sea urchin embryos and characterized it as a factor activating the ERK signals in these cells. Importantly, they demonstrated that KSR3, evolutionary conserved among non-chordate metazoans, is able to activate ERK signals independently of Ras using the HEK293T cell culture system. Detailed comparative analyses of Raf, KSR1/2, and KSR3 protein sequences from diverse phyla resulted in identification of novel activating mutations of human B-Raf and allowed the authors to conclude that KSR3 acts as an allosteric activator of Raf.

The study is executed beautifully and the manuscript is well written. It also highlights the importance of studying diverse animals. I have only a few comments, which the authors might think useful to improve the study:

1) Overexpression of wild type KSR3 in sea urchin embryos is not sufficient to activate ERK signals ectopically, which can be achieved only when PP2A-mediated dephosphorylation of KSR3 is accompanied. In contrast, Pmar1 overexpression is sufficient to ectopically activate ERK signals. This indicates that factors that modulate PP2A activity are among transcriptional targets downstream of the Pmar1/HesC double negative gate. Have the authors identified potential candidates for such factors in their RNA-seq screen?

2) The Pmar1/HesC/KSR3 regulatory cassette represents most likely an evolutionary novelty acquired in the "modern" sea urchin lineage. However, KSR3 is evolutionary conserved among non-chordate metazoans. As the authors discuss, it will be very interesting and important to look for KSR3 expression coupled with ERK activation pattern in diverse animals in order to address to what extent the KSR3-mediated ERK activation is deployed. While this type of studies is obviously out of the scope of the current study, I am interested in whether KSR3 is expressed during other developmental stages of sea urchin embryos and, if so, whether its expression is coupled with ERK activation. If the current study could provide an additional dataset showing that the KSR3-mediated ERK activation is operating not only during the "unique" micromere specification but also during other developmental processes likely to be conserved in echinoderms, it would add another "twist" to this nice study.

3) In Figure 10, the authors describe the KSR3-mediated ERK activation as "ancient". How do they know that it represents an ancient way to activate ERK? I would rather simply describe it as "alternative".

4) In the two discussion sections, "KSR3 as the signal that activates...." and "KSR3, the evolutionary history of....", a large part of statements is not referenced.

Reviewer #3 (Remarks to the Author):

The manuscript by Chessel et al focuses on the identification of mechanisms leading to RAS-independent ERK activation in early sea urchin development. The authors identify a novel transcript controlled by beta-Catenin signaling that codes for a novel KSR variant they authors call KSR3. KSR3 differs significantly from the classical paralogs KSR1 and KSR2 and constitutively activates RAF via dimerization. The identification of KSR3 is of particular interest because it also sheds light upon the requirements for BRAF activation in cancer. Moreover, this mechanism appears to be present in most non-chordate phyla, thus constituting a novel regulatory mechanism present in wide spectrum of species that adds an important aspect of MAPK signaling. Overall, the manuscript is of high quality and a major step forward towards to understanding of the various mechanisms leading to MAPK pathway activation. However, there are a few points that should be addressed until the manuscript can be published:

1. Is KSR3 only expressed during the early developmental stages until the SB? If this is the case, could the authors speculate why there would be a specific requirement for RAS-independent ERK activity in this situation and not in others?
2. KSR3 seems to have a highly remodeled ATP binding pocket. Since ATP binding was shown to be relevant for several activities of KSR1/KSR2, can KSR3 also bind ATP despite having a divergent ATP binding pocket?
3. Is KSR3 directly repressed by hesC or could another intermediate protein be involved? The most direct way to test this would be by performing ChIP assays with the KSR3 promoter.
4. Another question that arises is why it is necessary to dephosphorylate KSR3 when it is overexpressed to observe ERK activation. Conceptually, this would probably mean that endogenous KSR3, when induced by beta-Catenin signaling, is in fact dephosphorylated. Do pp2ab expression levels also change during the different developmental stages or when pmar1 is overexpressed? If not,

is there any evidence for pp2a activation upon formation of PMCs?

5. The experiments aimed at elucidating the mechanism of KSR3 in human 293T cells are definitely insightful, especially with regard to the role of the NtA region. However, some questions were not addressed. For instance, does ERK activation by KSR3 require membrane localization? Given the lack of a CC-SAM domain, one would expect that KSR3 acts in the cytoplasm. On the other hand, several amino acids within the CA3 of KSR1 also known to be required for interaction with the plasma membrane. The question here is whether KSR3 activates RAF at the plasma membrane or in the cytoplasm. In Fig. 10, the authors draw KSR3 associated with the membrane (is there any evidence?), whereas KSR1 was drawn in the cytoplasm, although evidence clearly indicates that RAS-dependent KSR1 activation involves its translocation to the membrane.

6. A S361E substitution in KSR3 creates a potent allosteric activator of BRAF due to bypassing of NtA phosphorylation. Would a similar mutation in the dimer interface of KSR1, which by itself does not activate ERK, also cause similar activation of ERK signaling? Moreover, would this mutation in KSR3 also override all regulatory mechanism in the embryo and cause an EMT phenotype?

7. Finally, KSR3 seems to play a role only in non-chordate phyla, but its general role (if any) is not clear. Are there any conserved transcription factor binding sites within the KSR3 promoter from different species? Is there any evidence that regulation of KSR3 expression by beta-Catenin signaling is conserved?

Minor comments:

- It could be a good idea to use a more specific title. In its current form, the readers will not be able to infer what the paper is about. Some reference to a novel KSR variant would certainly draw more attention.

- Fig. 8c does not have an untransfected control, which is present in all other figures.

REVIEWER COMMENTS

Reviewer #1 (Remarks to the Author):

This manuscript by Chessel et al. convincingly demonstrates that a constitutively active version of a gene related to vertebrate KSR1 and KSR2, termed KSR3, is necessary for the cell-autonomous, RTK-RAS independent, mechanism that activates p-ERK signaling in a subset of mesoderm cells downstream of the PMAR/HesC double negative gate. They go on to show that KSR3 is an evolutionarily conserved in most metazoans, excluding chordates. Then the authors go into detail to determine the role of the various domains of KSR3 and their interactions and activation of the RAF map kinase upstream of p-ERK. Finally, they use their data from the KSR3 mutations to identify novel putative mutations in human Raf that may have implications in human disease. Overall, this is a well written manuscript, and the experiments are thorough and well performed, and presented. This work will be of considerable interest to developmental biologist as well as scientists researching RTK/RAS/Raf signaling and its role in human disease and I recommend it for publication. However, there are a few issues that would improve the manuscript and should be addressed.

We thank the reviewer for this positive assessment of our manuscript.

Major issues:

- For figure 1e, control embryos should be shown for each experiment. We have added the controls for each experiment as requested.

- In the second section (Pg. 8, starting on line 196), the RNA-seq screen is not explained very well and could be confusing to readers. The authors should consider reworking the intro of this sections. Also, it could help if the authors created a better diagram of the RNA-seq screen in Fig. 2a instead of just showing the phenotypes of the perturbations?

We agree with the reviewer that the screen was not well explained. In particular, the reasons for using embryos devoid of beta catenin function as controls in the screen were **not** sufficiently explained. We have therefore rephrased the beginning of this section as follows: Having established that activation of ERK in mesodermal precursors requires the activity of the transcription factor Pmar1, we attempted to identify the “signal” that activates ERK by comparing the transcriptome of early embryos overexpressing *pmar1* to that of embryos lacking the activity of Pmar1. Since the function of Pmar1 cannot be easily blocked using morpholinos due to the expression of multiple transcripts isoforms generated from the *pmar1* locus, we used instead as a proxy for the Pmar1 loss-of-function condition, embryos animalized by overexpression of a dominant negative cadherin construct, a perturbation which inhibits the beta-catenin pathway, blocks *pmar1* expression and suppresses PMC specification and ERK activation (Fig. 2a)^{22,23}

- The KSR MO experiments require that specificity controls be performed. While they do use two morpholinos, there is no antibody available so they should do a standard rescue experiment with KSR mRNA co-injection. They use a splice blocking morpholino but don't show that it is blocking correct RNA processing using a simple PCR assay. They should perform this experiment.

We agree with the reviewer that it is important to control for the specificity of morpholinos. In their comprehensive review on the use of morpholinos, Eisen and Smith pointed that, “*for genes that are expressed in a restricted manner, obtaining a true rescue may be difficult or impossible since it might be necessary to drive expression of the gene in its correct spatial pattern at the correct level*”. Obviously, *ksr3*, which is expressed in a highly restricted manner in the PMCs precursors, is one such special example in which the rescue experiment may not be trivial to perform.

To test for the specificity of our morpholino, we attempted to rescue the phenotype caused by the translation blocking or splice blocking morpholino (in this case the lack of ERK activation in the PMC precursors at blastula stage) by injecting mRNA immune to the action of the morpholino (not containing the sequence recognized by the morpholino or containing a modified sequence). Both in the case of the translation blocking and splice blocking oligonucleotides, our attempts to rescue ERK activation in the PMC precursors by injection of wt *ksr3* mRNA into the egg did not work. Since wt *ksr3* mRNA does not show any detectable activity when overexpressed, we then compared the effect of injecting the morpholino oligonucleotide alone, with that caused by co-injection of the morpholino oligonucleotide together with mRNA encoding KSR3 S87,89A, which is the only KSR3 construct displaying an activity when overexpressed. While injection of the *ksr3* morpholino alone resulted in the absence of ERK activation in the PMC precursors, co-injection of the *ksr3* morpholino together with mRNA encoding the active KSR3 protein immune to the morpholino successfully restored ERK activation in the PMC precursors supporting the idea that the effect caused by our morpholino is specific to the inhibition of *ksr3* function.

As requested by the reviewer, we also controlled for the efficacy of our splice blocking morpholino. Our splice morpholino was designed to mask the exon4 intron 4 junction. (E4I4) Since, one common effect of such morpholino is exon skipping, we expected injection of the morpholino to cause deletion of exon 4 and to result in a frame shift 28 residues after the exon 3-intron 3 junction. However, RT-PCR analysis of embryos injected with the splice blocking morpholino revealed that the main consequence of the splice blocking oligonucleotide is not skipping of exon 4 but instead production of a shorter spliced variant. Sequencing of the PCR product revealed that this transcript was generated by utilization of a cryptic splice donor 36 bp upstream of the normal E4I4 junction. This abnormal splicing did not cause a frame shift in the coding sequence but it resulted in deletion of 12 amino acids of the P-loop of KSR3. The deleted protein sequence comprises isoleucine 321, glycine324 and lysine 331 residues that are highly conserved in almost all RAF and KSR proteins strongly suggesting that the protein synthesized from this abnormally spliced mRNA is inactive. A figure describing the molecular characterization of the effect of the splice morpholino has been added to the supplementary materials.

Finally, to further support the specificity of our morpholinos, we performed a synergy experiment. Each morpholino was injected alone at 0.25 mM, a dose that did not cause any phenotype, or in combination. While each morpholino at this dose did not elicit a phenotype and did not detectably affect ERK activation at the vegetal pole, injection of the combination of morpholinos resulted in development of embryos that failed to form PMCs and that lacked ERK signaling in the PMC precursors.

This novel piece data has been added to the supplementary Figure.

We hope that these novel results will satisfy the reviewers and that they will convince them that the phenotypes caused by the two morpholinos that we used are not off-targets effects of the morpholinos but result from a bona fide knockdown of *ksr3*.

- The conclusions stated for the paragraph starting on pg. 13, line 309 are overstated. The data indicate that KSR3 is necessary for RAS-independent activation phospho-ERK, not “absolutely” required and sufficient. To show sufficiency the authors should inject on of the constitutively active construct into a non-mesodermal blastomere and assess Erk activation and phenotypes. Thus, the authors should consider either softening the language or performing the experiments.

We agree with the reviewer that the claim that KSR3 is “absolutely” required was unnecessarily emphasized. We have therefore toned down our conclusion and deleted “absolutely”.

Regarding the second statement about sufficiency, and our conclusion that “*transcription and translation of ksr3 mRNA are necessary and sufficient steps for the RAS-independent activation*”. The reviewer is correct to write: “to show sufficiency the authors should inject the constitutively active construct...and assess ERK activation and phenotypes”. However, we are a little confused since is not it precisely the experiment that we had done in Fig. 3b (panels on the right)? In this experiment, we had shown that overexpression of a constitutively active *ksr3* causes massive and ectopic ERK activation in all cells of the embryo. This being said, the reviewer is correct to point that the claim about sufficiency should be used with caution since it is overexpression of the non phosphorylatable construct that is sufficient to activate ERK. Therefore, strictly speaking, transcription and translation of KSR3 may not be sufficient to activate ERK signalling and dephosphorylation of KSR3 is likely another step required for activation of ERK. This is why we have removed the notion of sufficiency in the sentence as suggested by the reviewer.

- The functional experiments showing that mutating RAF dimerization sites in KSR1 and KSR3 are interesting, but the authors overstate their conclusions from these experiments. At this point in the story these experiments do not prove that KSR3 dimerizes with RAF and it is not clear how these results suggest that “KSR3 prevents allosteric transactivation of Raf by KSR3”?

We are a little confused by the comment of the reviewer since the citation used is not accurate. We never said that “KSR3 prevents allosteric transactivation of Raf by KSR3”. We wrote instead that: “This **suggests** that the **R364H** mutant of KSR3 is not only a loss-of-function mutant but that it has a potent dominant negative activity that blocks the function of endogenous KSR3 and prevents allosteric transactivation of RAF by KSR3”. Indeed, it is the dominant negative construct, R364H mutant of KSR3, not the KSR3 wt, that prevents transactivation of RAF. We think that our statement was therefore correct. Also, we would like to bring the attention of the reviewer that we use the verb “suggest” in our statement which is not a particularly strong verb. Now, from the comment of the reviewer, we understand that his main point here is that he/she thinks that it is too early in the paper to talk about the allosteric transactivation of BRAF by KSR3. Although we have mentioned in the Introduction that KSR factors work as allosteric activators of RAF, we understand the point of the reviewer and we have deleted the end of the sentence that was saying:” and prevents allosteric transactivation of RAF by KSR3”, keeping only the idea that the dominant R364H mutant of KSR3 blocks the function of endogenous KSR3.

The authors should consider placing these experiments in Fig. 9 after they biochemical experiments showing dimerization between KSR3 and RAF. These functional in vivo results would strengthen the argument there instead of seeming out of place where they currently stand.

We have considered the suggestion of the reviewer but unfortunately we have a different opinion on that matter. We feel that showing the phenotypes of the R364H mutant of KSR3 in Fig. 3 is important since it reinforces the demonstration that KSR3 function is required for ERK activation and for EMT of the mesodermal precursors.

Also, we show that KSR3 co-immunoprecipitates with BRAF, in Fig 6c, i.e. well before Fig. 9. Similarly, we show that dimerization defective KSR3 mutants fail to activate ERK signalling in three different species in Fig. 6. Finally we also show well before Fig. 9 that the dominant negative CRAF constructs block the activation of ERK by KSR3 in these three different species strongly suggesting that KSR3 act through RAF to activate ERK signalling.

Finally, we would like to draw the attention of the reviewer on the fact that the main point of Fig. 9 is not to demonstrate that KSR3 dimerizes with BRAF (this was already shown in Fig. 6) but to show that the constitutively active KSR3 S361E and B-RAF R506E mutants activate ERK signalling both by promoting dimerization and by strengthening transactivation of the B-RAF receivers.

Also, these experiments do not show the WT KSR3 overexpression phenotypes when it seems they should be included with these experiments and then a sentence or two about why they don't elicit a phenotype. Also, do the authors address why overexpressing KSR3 in sea urchins doesn't increase p-Erk whereas it does increase p-Erk in human cell lines (Fig. 6a)? We are a little confused by the comment of the reviewer on the absence of *ksr3* overexpression phenotypes. We agree with the reviewer that it is important to show *ksr3* overexpression phenotypes and this is why we had, indeed, included WT KSR3 overexpression phenotypes in Fig3c (second column of panels) of the manuscript.

Regarding the suggestion to add a sentence or two about why they don't elicit a phenotype, it seems to us that this is precisely we had done in the section titled: **KSR3 requires dephosphorylation on the N-terminus to be active**, which started as follows:

“Surprisingly, overexpression of wild-type *ksr3* did not cause massive ectopic activation of ERK or massive EMT. Because the activity of KSR factors is negatively regulated by the binding of 14-3-3 proteins to the phosphorylated serine at the N-terminal region (serine 259)^{5,33-36}, and positively regulated by dephosphorylation by PP2A, this raised the possibility that sea urchin KSR3 activity is repressed by a similar mechanism. Indeed, overexpression of a mutant form of KSR3 in which the two serine residues (Ser84, Ser87) (Fig. 3b) in position equivalent to Ser259 of human B-RAF were substituted by alanine residues caused a massive ectopic activation of ERK (Fig. 3b)

- The entire sections entitled “The ability of KSR3 proteins to activate ERK signaling....” on page 21 line 533 as well as the section entitled “KSR3 S361E....” on page 30 are hard to follow as written. A particular problem is that it is difficult to match the various constructs that they talk about to the experiments shown in Figure 6. They should better define the various constructs in the manuscript.

We apologize for the lack of clarity. There were two reasons why it was difficult to match the constructs described in the text and the experiments of Fig. 6. The first is that the text was referring to Fig. 6 g,h while only Fig. 6g was relevant and the second is that the names of the two NtA mutants described in the text was different from the names of the mutants described in Fig. 6. We have deleted the reference to Fig. 6h and homogenized the names of the constructs in the text and in the Figure. We thank the reviewer for pointing this to us.

Also, the authors place a KSR3 dimerization mutant experiment in figure 6e. Why not place this in Figure 9. Again, this seems out of place like the previous dimerization experiment in Fig 3.

As explained above, we prefer to present the biochemical experiments showing the critical roles of the dimerization defective Arginine mutants for all three species in Fig. 6 as well as the experiments showing inhibition of ERK activation by KSR3 from all three species following overexpression of a dominant negative CRAF construct since we want to convince the reader that what we describe is not a peculiarity of the sea urchin but a property shared by several KSR3 family members.

- The same critique could be applied, although to a lesser extent, for the other western blot/biochemical interactions sections. All these experiments are very well-done experiments, but the data are extremely dense, and it would be better on readers if the authors provided more context and made it clear which constructs that are talking about in the main body of the manuscript. For instance, the authors could indicate the abbreviation used in the figures in parentheses after they describe the mutant in the text. A diagram of the various mutations the use in each of these sections could be added to these figures for easier reference instead of going back and forth to figure 2 and figures 5-9. It's hard work.

We understand the critique of the reviewer. There are a lot of constructs and it is not always easy to match the text and the figures. To facilitate the task of the reader, we have followed the suggestion of the reviewer and added a scheme describing the positions of the different mutations that we used. We now indicate in the main text the name of the construct used in the figures. For example: (construct P1 KSR3 VTA^{insertion 345}) (Fig. 8b)). This should facilitate the understanding of these biochemical and western blot experiments. We thank the reviewer for pointing that.

- The discussion is interesting, but very long. The authors should consider reworking this section so that it is more succinct.

We are pleased that the reviewer finds the Discussion interesting. According to the reviewer however, the Discussion section may appear a little long to the reader. We have taken into account the comment of the reviewer and pruned the Discussion in several places. For example, in the first section of the Discussion we have deleted several sentences that were more a repetition of results than a real discussion.

Minor issues:

- There are many grammatical errors throughout the manuscript, including several run-on

sentences (e.g., the sentence on page 35, lines 817-832). Also, it is hard to determine where paragraphs start/end because there are no indentations at the beginning of paragraphs or spaces between them. The authors should perform a careful edit of the writing before resubmission.

We apologize for the grammatical errors. We have carefully edited the manuscript to remove these small errors and added spaces after paragraphs to better recognize them.

- beta-catenin should be capitalized if talking about the protein or use the Greek Beta. We have followed the reviewer's recommendation.

- Image in Fig1C for VEB *ets1* is not clear. Is *ets1* expressed ubiquitously at this stage? This is a good point raised by the reviewer. Yes, *ets1* is expressed maternally and ubiquitously and the maternal expression partially obscures the zygotic expression in the PMC precursors. This is now mentioned in the legend of Fig. 1.

- The writing in the section entitled "KSR is required for activation of ERK..." on page 13 is difficult to follow. In the first and second paragraphs of the section the authors should consider providing more context for the experiments. As is, these paragraphs read as a list of what they did.

We have provided more context in this paragraph as requested.

- Were the *HesC*, *Ets1*, *Alx1* and *KSR1* morpholinos previously characterized? If so, then these morpholinos should be referenced.

To our knowledge, none of these morpholinos have been used in *Paracentrotus* but *hesC*, *ets1* and *alx1* have been used in other species. We have provided references for these three morpholinos.

- Line 312 p13 the sentence states "We confirmed these results with a second morpholino..." needs a figure indication showing the phenotype of these morphants.

We have provided a figure indication as requested.

Reviewer #2 (Remarks to the Author):

In this study, the authors first identified KSR3 as a transcriptional target of Pmar1 (via a double negative gating) in micromeres of sea urchin embryos and characterized it as a factor activating the ERK signals in these cells. Importantly, they demonstrated that KSR3, evolutionary conserved among non-chordate metazoans, is able to activate ERK signals independently of Ras using the HEK293T cell culture system. Detailed comparative analyses of Raf, KSR1/2, and KSR3 protein sequences from diverse phyla resulted in identification of novel activating mutations of human B-Raf and allowed the authors to conclude that KSR3 acts as an allosteric activator of Raf.

The study is executed beautifully and the manuscript is well written. It also highlights the importance of studying diverse animals.

We thank the reviewer for this positive judgment on our study.

I have only a few comments, which the authors might think useful to improve the study:

1) Overexpression of wild type KSR3 in sea urchin embryos is not sufficient to activate ERK signals ectopically, which can be achieved only when PP2A-mediated dephosphorylation of KSR3 is accompanied. In contrast, Pmar1 overexpression is sufficient to ectopically activate ERK signals. This indicates that factors that modulate PP2A activity are among transcriptional targets downstream of the Pmar1/HesC double negative gate. Have the authors identified potential candidates for such factors in their RNA-seq screen?

The reviewer raises a very interesting point that was also raised by the other two reviewers. He/she is perfectly correct to notice that overexpression of *ksr3* is not sufficient to cause ectopic activation of ERK while co-overexpression of *ksr3* with *pp2a* is sufficient to trigger massive activation of ERK in non-PMC cells as does overexpression of a non phosphorylatable form of *ksr3*. As pointed by the reviewer, the unavoidable conclusion is that factors that modulate PP2A activity are likely among transcriptional targets of the Pmar1/HesC double negative gate. This is exciting since it suggests that not only Pmar1 activates *ksr3* expression but that it may also activates the expression of a key regulator of KSR3 activity. Following the reviewer's suggestion, we have carefully examined the results of the RNA-seq screen looking for potential candidates. Unfortunately, we did not find genes encoding regulators of PP2A in the list of genes upregulated following Pmar1 overexpression. We then went on and cloned the four different genes encoding regulatory subunits of PP2A present in the sea urchin genome (PP2AR 56 kDa Delta, PP2AR 56kDa epsilon, PP2AR 3C, PP2AR 55kDa Delta). For each one we made two different in situ probes and analysed the expression pattern. Although several of these genes showed expression in the delaminated PMCs at mesenchyme blastula stage, none of these genes showed a restricted expression in precursors of the PMCs at blastula stages. Therefore, although the idea of the reviewer is very attractive, identifying the factor that is regulated by Pmar1 and that controls the activity of KSR3 is presently challenging.

2) The Pmar1/HesC/KSR3 regulatory cassette represents most likely an evolutionary novelty acquired in the "modern" sea urchin lineage. However, KSR3 is evolutionary conserved among non-chordate metazoans. As the authors discuss, it will be very interesting and important to look for KSR3 expression coupled with ERK activation pattern in diverse animals in order to address to what extent the KSR3-mediated ERK activation is deployed. While this type of studies is obviously out of the scope of the current study, I am interested in whether KSR3 is expressed during other developmental stages of sea urchin embryos and, if so, whether its expression is coupled with ERK activation. If the current study could provide an additional dataset showing that the KSR3-mediated ERK activation is operating not only during the "unique" micromere specification but also during other developmental processes likely to be conserved in echinoderms, it would add another "twist" to this nice study.

We understand and share the curiosity of the reviewer . Just like the reviewer, we wondered whether, in addition to regulating specification of the PMCs, KSR3 may be involved in other developmental processes. To satisfy the reviewer's curiosity we first looked at RNA-seq data and found that in addition to the peak of *ksr3* expression at blastula stage, *ksr3* is re-expressed after the pluteus stages. In particular, we have detected significant expression in late 4-arms and in 6 and 8-arms pluteus larvae, strongly suggesting that KSR3 does regulate other ERK-dependent processes just like it does during PMC specification.

We then went on and using a much more sensitive *ksr3* probe, we repeated in situ hybridizations at stages that were not analyzed in our previous study. At the beginning of gastrulation, we were able to detect *ksr3* expression in the newly invaginated archenteron, as well as in neurons of the animal pole that likely correspond to serotonergic neurons. At later stages, we detected *ksr3* transcripts in the roof of the archenteron, in the apical plate and in a single cells within the ectoderm and archenteron. Finally, we detected *ksr3* expression in the developing appendages of the adult rudiment. We also performed p-ERK immunostaining at these stages and found that the pattern of *ksr3* expression is largely overlapping with the pattern of ERK activation, strongly suggesting that KSR3-mediated ERK activation is not limited to the PMCs but that it also occurs at different developmental stages and in different tissues. This novel piece of data has been added to the paper and now constitutes the main part of Fig. 3. We hope that this novel piece of data will satisfy the reviewer's curiosity.

3) In Figure 10, the authors describe the KSR3-mediated ERK activation as "ancient". How do they know that it represents an ancient way to activate ERK? I would rather simply describe it as "alternative".

We wrote that the mechanism of KSR3-mediated ERK activation is ancient because our phylogeny analysis indicates that *ksr3* genes likely appeared before the emergence of metazoa. In this respect KSR3 mediated activation of ERK is just as ancient as the canonical ERK activation pathway and KSR3 is present in the genomes and expressed in the transcriptomes of placozoa and cnidarians, two phyla that are have emerged very early in the evolutionary history of animals.

4) In the two discussion sections, "KSR3 as the signal that activates...." and "KSR3, the evolutionary history of....", a large part of statements is not referenced.

Reviewer #3 (Remarks to the Author):

The manuscript by Chessel et al focuses on the identification of mechanisms leading to RAS-independent ERK activation in early sea urchin development. The authors identify a novel transcript controlled by beta-Catenin signaling that codes for a novel KSR variant they authors call KSR3. KSR3 differs significantly from the classical paralogs KSR1 and KSR2 and constitutively activates RAF via dimerization. The identification of KSR3 is of particular interest because it also sheds light upon the requirements for BRAF activation in cancer. Moreover, this mechanism appears to be present in most non-chordate phyla, thus constituting a novel regulatory mechanism present in wide spectrum of species that adds an important aspect of MAPK signaling. Overall, the manuscript is of high quality and a major step forward towards to understanding of the various mechanisms leading to MAPK pathway activation.

We thank the reviewer for this positive assessment of our work.

However, there are a few points that should be addressed until the manuscript can be published:

1. Is KSR3 only expressed during the early developmental stages until the SB? If this is the

case, could the authors speculate why there would be a specific requirement for RAS-independent ERK activity in this situation and not in others?

The question raised by the reviewer is very interesting and was shared with reviewer 2. Regarding the first question, as explained above, we have provided a novel in situ hybridization dataset showing that KSR3 is also expressed during gastrulation in various cell types and during late larval development in the adult rudiment. This piece of data is now included in Fig. 3. Regarding the second question and why there would be a specific requirement for RAS-independent ERK activity in this situation and not in others, we have put forward the idea that activation of ERK signaling by the transcriptional activation of *ksr3* may be a widespread mechanism used in the course of evolution for the co-option of the ERK pathway in novel locations during development, as shown for example in the case of activation of ERK in the skeletogenic precursors in euechinoids. KSR3-mediated activation of ERK may combine several features that may make it particularly well adapted for the co-option of the ERK pathway in a novel tissue: it allows strong and cell autonomous activation of ERK in sharply delimited cell populations or even in single cells and it is relatively parsimonious in term of number of regulatory mutations needed since it only requires the *ksr3* gene to be placed under the control of a lineage specific transcription factor. These ideas are now mentioned in the Discussion section.

2. KSR3 seems to have a highly remodeled ATP binding pocket. Since ATP binding was shown to be relevant for several activities of KSR1/KSR2, can KSR3 also bind ATP despite having a divergent ATP binding pocket?

The reviewer raises another interesting point. He/she is correct to point that KSR3 possesses a highly remodeled ATP binding site. Both the Ploop, which is crucial for coordination of the non-transferable phosphates of ATP, and key residues that constitute the nucleotide binding pocket (residues C532, W531, L514, and A481 in human B-RAF) are highly remodeled (only V471 and T529 are conserved). Although for these reasons we think that KSR3 does not bind ATP, this cannot be formally excluded and therefore the reviewer question about ATP-binding of KSR3 is justified. To answer the question of the reviewer we have made and expressed a mutant of the ATP binding pocket that should not be able to bind ATP. This mutation is the equivalent of the A587F mutant of KSR1 described by Hu and coll. (Mutation that blocks ATP binding creates a pseudokinase stabilizing the scaffolding function of kinase suppressor of Ras, CRAF and BRAF and equivalent to the A703F mutant of *Drosophila* KSR (see also Supporting Information Hu et al. 10.1073/pnas.1102554108) and both mutants have been shown to lack ATP binding and ERK activation.

We found that, unlike the vertebrate and fly KSR ATP-binding site mutants, which have lost the ability to activate ERK, the sea urchin L337F KSR3 mutant fully conserved its ability of to activate ERK signaling. We therefore conclude that it is very unlikely that KSR3 binds ATP.

3. Is *ksr3* directly repressed by HesC or could another intermediate protein be involved? The most direct way to test this would be by performing ChIP assays with the KSR3 promoter. This is another interesting question raised by the reviewer. Formally, we cannot exclude that in addition to HesC, there could be another ubiquitous factor repressing the *ksr3* gene; However, it seems to us that if this were the case, there would have to be not one but two repressors acting after HesC since the final goal is activation of *ksr3*. The circuit would therefore be:

Beta catenin → Pmar1 --- <HesC --- <repressorX --- <repressorY --- <*ksr3*

The end result would be expression of the repressorX in the mesoderm and expression of repressor Y in the ectoderm which is what we want.

With only one repressor after HesC the circuit would be:

Beta catenin → Pmar1 --- <HesC --- <repressorX --- <*ksr3*

and it would lead to expression of *ksr3* in the ectoderm, not to repression of *ksr3*.

Although the circuit with 2 more repressors is formally possible we think that it is unlikely since our novel in situ hybridization analysis shows that *ksr3* starts to be expressed at the 60 cell stage while the earliest expression of Pmar1 is detected at the 32 cell stage. There seems to be very little time to accommodate the expression of an additional repressor X between the expression of Pmar1 and the expression of *ksr3*.

We agree that ChIP assays with the *ksr3* promoter may be a technique that could provide information on that question. However and unfortunately, we don't have an antibody specific to the sea urchin HesC factor to perform ChIP assays. We also feel that although the question is interesting, it is not crucial for the point that we want to make in this manuscript which is the discovery of an alternative mode of ERK activation by a family of constitutively active KSR3 factors.

4. Another question that arises is why it is necessary to dephosphorylate KSR3 when it is overexpressed to observe ERK activation. Conceptually, this would probably mean that endogenous KSR3, when induced by beta-Catenin signaling, is in fact dephosphorylated. Do pp2ab expression levels also change during the different developmental stages or when pmar1 is overexpressed? If not, is there any evidence for pp2a activation upon formation of PMCs? Again, this is an excellent remark. This question regarding the dependence of KSR3 on dephosphorylation in order to be active was raised by all three reviewers and we thank all of them not only for asking these questions but for suggesting possible explanations for these observations. As explained above, and as suggested specifically by reviewer 2 and reviewer 3, one possible explanation for the requirement to dephosphorylate KSR3 for this factor to be able to activate ERK could be that Pmar1 may control the expression of a phosphatase that would therefore be co-expressed with *ksr3*. Unfortunately, we have not identified candidate factors that would fulfill this role and that would be expressed specifically in the PMCs downstream of Pmar1.

5. The experiments aimed at elucidating the mechanism of KSR3 in human 293T cells are definitely insightful, especially with regard to the role of the NtA region. However, some questions were not addressed. For instance, does ERK activation by KSR3 require membrane localization? Given the lack of a CC-SAM domain, one would expect that KSR3 acts in the cytoplasm. On the other hand, several amino acids within the CA3 of KSR1 also known to be required for interaction with the plasma membrane. The question here is whether KSR3 activates RAF at the plasma membrane or in the cytoplasm. In Fig. 10, the authors draw KSR3 associated with the membrane (is there any evidence?), whereas KSR1 was drawn in the cytoplasm, although evidence clearly indicates that RAS-dependent KSR1 activation involves its translocation to the membrane.

Again, the question raised by the reviewer about the subcellular localization of KSR3 is potentially interesting and we are sorry for not having satisfied his/her curiosity by addressing this question in detail in our study. We thank the reviewer for acknowledging that we have, however, addressed several other questions more thoroughly.

As the reviewer pointed it, the CA3 also known as C1 or Cystein Rich Domain (CRD) is a domain thought to interact with phospholipids. The CRD domains of B-RAF and KSR3 may therefore be sufficient to drive membrane localization of the KSR3/B-RAF complex. Although we lack direct evidence so far supporting this hypothesis, we think that we may have provided indirect evidence suggesting that KSR3 and B-RAF are indeed recruited to the membrane during the KSR3-mediated ERK activation cycle. The first evidence is that the CRD domain appears to be extremely conserved since there is no KSR or RAF factors, to our knowledge, that does not contain a CRD domain, and the second is that we have shown that a deletion that removes the first 307 amino acids of KSR3, and therefore that removes the CRD of KSR3, has a severe effect on its ability to activate ERK signaling (Supplementary Figure 7) despite the fact that this deletion removes the N-terminal inhibitory 14-3-3 binding site. A possible explanation for the effect of this deletion is therefore that removing the CRD domain may impair the membrane localization of KSR3. To test this idea more directly, we mutated two conserved cysteines within the C1 domain. Replacing cysteines 37 and 40 of KSR3 by serine strongly reduced the ability of KSR3 to activate ERK (Fig. 8d construct PL KSR3 C37/40S) reinforcing the idea that integrity of the C1 domain is important for KSR3 function possibly through membrane localization.

The reviewer asks why, in the scheme of Fig. 11, KSR1/2 is depicted as a cytoplasmic protein and points that there is evidence that during RAS-dependent ERK activation KSR1/2 is translocated to the membrane. We completely agree with the reviewer and thank him/her for pointing that to us. We have therefore modified the scheme of Fig. 11 and now KSR1/2 appears localized to the membrane during canonical RAS-dependent ERK signaling.

6. A S361E substitution in KSR3 creates a potent allosteric activator of BRAF due to bypassing of NtA phosphorylation. Would a similar mutation in the dimer interface of KSR1, which by itself does not activate ERK, also cause similar activation of ERK signaling? Moreover, would this mutation in KSR3 also override all regulatory mechanism in the embryo and cause an EMT phenotype?

Again this is an interesting point that the reviewer raised and to answer this question we made the corresponding KSR1 mutant and tested its ability to activate ERK. Unfortunately, mutation of R608 of KSR1 into E did not increase its ability to activate ERK suggesting that other determinants control its activity.

7. Finally, KSR3 seems to play a role only in non-chordate phyla, but its general role (if any) is not clear. Are there any conserved transcription factor binding sites within the KSR3 promoter from different species? Is there any evidence that regulation of KSR3 expression by beta-Catenin signaling is conserved?

We think that the general role of KSR3 in non-chordate phyla is to activate ERK signaling in territories with sharp boundaries and in single cells. Regarding the question as to whether there are conserved binding sites for transcription factors within the promoter of KSR3 in different species, we have not yet looked at this question but we think that this is possible. For example if KSR3 has a conserved expression in neurons in different species, then it is conceivable that this expression may be driven by conserved transcription factors involved in specification of these cells. We think however, that this analysis is beyond the scope of this paper and would prefer to keep it for future studies.

Minor comments:

- It could be a good idea to use a more specific title. In its current form, the readers will not be able to infer what the paper is about. Some reference to a novel KSR variant would certainly draw more attention.

We have followed the suggestion of the reviewer and replaced the title by a more specific one:

RAS-independent ERK activation by KSR3, a member of a novel family of constitutively active Kinase Suppressor of Ras factors present in non-chordate metazoa

- Fig. 8c does not have an untransfected control, which is present in all other figures. We added the missing control to Figure 9.

REVIEWERS' COMMENTS

Reviewer #1 (Remarks to the Author):

The authors did a nice job addressing my concerns and I recommend the manuscript for publication.

Reviewer #2 (Remarks to the Author):

The authors addressed all my concerns. The study is beautifully executed and the manuscript reads well. I have only a few comments:

- 1) Fig 3c: The two prism larva in the bottom row (lv and sv) don't look like they are at the right stage. Same for the early pluteus larvae in the lateral view (top row). I might be wrong.
- 2) It will be nice for readers if the authors can indicate the existence of Supplementary table 2 somewhere in the main text.
- 3) Fig 10 was missing.

Reviewer #3 (Remarks to the Author):

The authors have addressed my comments satisfactorily.

REVIEWER COMMENTS

Reviewer #1 (Remarks to the Author):

The authors did a nice job addressing my concerns and I recommend the manuscript for publication.

Reviewer #2 (Remarks to the Author):

The authors addressed all my concerns. The study is beautifully executed and the manuscript reads well. I have only a few comments:

- 1) Fig 3c: The two prism larva in the bottom row (lv and sv) don't look like they are at the right stage. Same for the early pluteus larvae in the lateral view (top row). I might be wrong.*
- 2) It will be nice for readers if the authors can indicate the existence of Supplementary table 2 somewhere in the main text.*
- 3) Fig 10 was missing.*

Reviewer #3 (Remarks to the Author):

The authors have addressed my comments satisfactorily.

We thank all three reviewers for their positive assessment of our revised version of the manuscript.

Regarding the comment of reviewer 2 on Fig. 3c, we confirm that the stages indicated are correct.

Also we have followed the advice of the reviewer and added sentences indicating the existence of Supplementary Table 1 and 2.